# $O_2$:$CO_2$ exchange ratio for net turbulent flux observed in an Urban Area of Tokyo, Japan and its application to an evaluation of anthropogenic $CO_2$ emissions

Shigeyuki Ishidoya[1], Hirofumi Sugawara[2], Yukio Terao[3], Naoki Kaneyasu[1],
Nobuyuki Aoki[1], Kazuhiro Tsuboi[4], and Hiroaki Kondo[1]

[1]National Institute of Advanced Industrial Science and Technology (AIST), Tsukuba 305-8569, Japan
[2]Department of Earth and Ocean Sciences, National Defense Academy of Japan, Yokosuka 239-8686, Japan
[3]National Institute for Environmental Studies, Tsukuba 305-8506, Japan
[4]Meteorological Research Institute, Tsukuba 305-0052, Japan

*Correspondence to*: Shigeyuki Ishidoya (s-ishidoya@aist.go.jp)

**Abstract.** In order to examine $O_2$ consumption and $CO_2$ emission in a megacity, continuous observations of atmospheric $O_2$ and $CO_2$ concentrations, along with $CO_2$ flux, have been carried out simultaneously since March 2016 at the Yoyogi (YYG) site located in the middle of Tokyo, Japan. An average $O_2$:$CO_2$ exchange ratio for net turbulent $O_2$ and $CO_2$ fluxes ($OR_F$) between the urban area and the overlying atmosphere was obtained based on an aerodynamic method using the observed $O_2$ and $CO_2$ concentrations. The yearly mean $OR_F$ was found to be 1.62, falling within the range of the average OR values of liquid and gas fuels, and the annual average daily mean $O_2$ flux at YYG was estimated to be -16.3 μmol m$^{-2}$s$^{-1}$ based on the $OR_F$ and $CO_2$ flux. By using the observed $OR_F$ and $CO_2$ flux, along with the inventory-based $CO_2$ emission from human respiration, we estimated the average diurnal cycles of $CO_2$ fluxes from gas and liquid fuels consumption separately for each season. Both the estimated and the inventory-based $CO_2$ fluxes from gas fuels consumption showed average diurnal cycles with two peaks, one in the morning and another one in the evening; however, the evening peak of the inventory-based gas consumption was much larger than that estimated from the $CO_2$ flux. This can explain the discrepancy between the observed and the inventory-based total $CO_2$ flux at YYG. Therefore, simultaneous observations of $OR_F$ and $CO_2$ flux are useful in validating $CO_2$ emission inventories from statistical data.

## 1. Introduction

Precise observation of the atmospheric $O_2$ concentration ($O_2$/$N_2$ ratio) has been carried out since the early 1990s to elucidate the global $CO_2$ cycle (Keeling and Shertz, 1992). The approach is based on the -$O_2$:$CO_2$ exchange ratios (Oxidative Ratio; $OR = -\Delta O_2 \Delta CO_2^{-1}$ mol mol$^{-1}$) for the terrestrial biospheric activities and fossil fuel combustion. The OR value of 1.1 has been used widely for the terrestrial biospheric $O_2$ and $CO_2$ fluxes (Severinghaus, 1995). On the other hand, the OR of 1.95 for gaseous fuels, 1.44 for oil and other liquid fuels, and 1.17 for coal or solid fuels are usually used (Keeling, 1988).

Therefore, OR is a useful indicator for cause(s) of the observed variations in the atmospheric $O_2$ and $CO_2$ concentrations. The atmospheric $CO_2$ concentration has been observed not only at remote sites such as Mauna Loa (19.5 °N, 155.6 °W), Hawaii, U.S.A. to capture a baseline variation in the background air (e.g. Keeling et al., 2011) but also recently in urban areas to estimate $CO_2$ emissions locally from fossil fuel combustion (e.g. Mitchell et al., 2018; Sargent et al., 2018). For the latter purpose, simultaneous observations of the atmospheric $O_2$ and $CO_2$ concentrations should provide important insight into validating the inventory-based $CO_2$ emissions from gaseous, liquid and solid fuels. Steinbach et al. (2011) estimated a global dataset of spatial and temporal variations of OR for the fossil fuel combustion using the EDGAR (Emission Database for Global Atmospheric Research) inventory and fossil fuel consumption data from the UN energy statistics. The statistically estimated OR should be validated by observed OR, however observations of the atmospheric $O_2$ concentration in urban areas are still limited (e.g. van der Laan et al., 2014; Goto et al., 2013a). Moreover, simultaneous observations of the OR and $CO_2$ flux between an urban area and the overlying atmosphere have never been reported before. Observations of the $CO_2$ flux have been carried out at various urban stations such as, London, UK (Ward et al., 2013), Mexico City, Mexico (Velasco et al., 2009), Beijing, China (Song and Wang, 2012), and Tokyo, Japan (Hirano et al., 2015), allowing us to observe urban $CO_2$ emission directly in the flux footprint. Therefore, if the OR for the net turbulent $O_2$ and $CO_2$ fluxes (hereafter referred to as "$OR_F$") can be observed, then such information can be used as a useful constraint for evaluating the contributions of the gaseous, liquid, and solid fuels, and the terrestrial biospheric activities to the observed $CO_2$ flux. From the measurements, it also becomes possible to observe the urban $O_2$ flux by multiplying the $CO_2$ flux by $OR_F$.

In this paper, we first present the simultaneous observational results of the $O_2$ and $CO_2$ concentrations and the $CO_2$ flux in the urban area of Tokyo, Japan. From a relationship between the vertical gradients of the observed $O_2$ and $CO_2$ concentrations, we derive $OR_F$ based on an aerodynamic method (Yamamoto et al., 1999). The present paper follows Ishidoya et al. (2015) who reported $OR_F$ for the $O_2$ and $CO_2$ fluxes between a forest canopy and the overlying atmosphere. We also compare the observed $OR_F$ with the OR value of the overlying atmosphere above the urban canopy (hereafter referred to as "$OR_{atm}$") to highlight the characteristics of the $O_2$ and $CO_2$ exchange processes in the urban canopy air at the YYG site. Finally, we estimate the average diurnal cycles of $CO_2$ fluxes from gas and liquid fuels consumption separately by using the $OR_F$, $CO_2$ flux, and inventory-based $CO_2$ emission from human respiration, in order to validate the inventory-based $CO_2$ emissions from gas consumption and traffic.

## 2. Experimental procedures

### 2.1 Site description

In order to observe the atmospheric $O_2$ and $CO_2$ concentrations and $CO_2$ flux between the urban area and the overlying atmosphere, the instruments were installed on a roof-top tower of Tokai University (52 m above ground, 25 m above roof) at Yoyogi (YYG; 35.66°N, 139.68°E), Tokyo, Japan. The YYG site is a mid-rise residential area and located in the northern

part of Shibuya ward, Tokyo. Figure 1 shows the location of the YYG site and the flux footprints averaged for summer and winter runs, calculated by the model of Neftel et al. (2008). The main land-cover around the site is characterized by low- to mid-rise residential buildings with a mean height of 9 m. The population density in this area is 16,600 persons $km^{-2}$. At the
YYG site, prevailing wind is from SW in the summer and NW in the winter. The flux footprint includes vegetated area of 9% in the summer and 2% in the winter, reflecting seasonal changes in the wind direction.

## 2.2 Continuous measurements of the atmospheric $O_2$ and $CO_2$ concentrations and $CO_2$ flux

Observations of the atmospheric $O_2$ and $CO_2$ concentrations have been carried out at the YYG site using a continuous measurement system employing a paramagnetic $O_2$ analyzer (POM-6E, Japan Air Liquid) and a non-dispersive infrared $CO_2$
analyzer (NDIR; Li-820, LI-COR) since March 2016. The $O_2$ concentration is reported as the $O_2/N_2$ ratio in per meg:

$$\delta\left(O_2/N_2\right) = \left[\frac{\left(O_2/N_2\right)_{sample}}{\left(O_2/N_2\right)_{standard}} - 1\right] \times 10^6 \quad (eq.1)$$

where the subscripts 'sample' and 'standard' indicate the sample air and the standard gas, respectively. Because $O_2$ is about 20.94 % of air by volume (Tohjima et al., 2005a), the addition of 1 μmol of $O_2$ to 1 mol of dry air increases $\delta(O_2/N_2)$ by 4.8 per meg (=1/0.2094). If $CO_2$ were to be converted one-for-one into $O_2$, this would cause an increase of 4.8 per meg of
$\delta(O_2/N_2)$, equivalent to an increase of 1 μmol $mol^{-1}$ of $O_2$ for each 1 μmol $mol^{-1}$ decrease in $CO_2$. Therefore, the ratio of 4.8 per meg/μmol $mol^{-1}$ was used to convert the observed $\delta(O_2/N_2)$ to $O_2$ concentration relative to an arbitrary reference point. In this study, $\delta(O_2/N_2)$ values of each air sample were measured with the paramagnetic analyzer using working standard air that was measured against our primary standard air (Cylinder No. CRC00045; AIST-scale) using a mass spectrometer (Thermo Scientific Delta-V) (Ishidoya and Murayama, 2014).
Sample air was taken at the tower heights of 52 m and 37 m using a diaphragm pump at a flow rate higher than 10 L $min^{-1}$ to prevent thermally-diffusive fractionation of air molecules at the air intake (Blaine et al., 2006). Then, a large portion of the air is exhausted from the buffer, with the remaining air allowed to flow into the analyzers from the center of the buffer. It is then sent to an electric cooling unit with a water trap cooled to −80°C at a flow rate of 100 mL $min^{-1}$, with the pressure stabilized to 0.1 Pa and measured for 10 minutes at each height (1-cycle measurements). The method to sample a small
subset of air from high flow rate is similar to those used in Goto at el. (2013b), and we have confirmed that the atmospheric $\delta(O_2/N_2)$ values observed by the measurement system agree well with those obtained from independent continuous measurements of $\delta(O_2/N_2)$ using the mass spectrometer (see Fig. 4 in Ishidoya et al., 2017). After 9 cycles of measurements (5 and 4 cycles for 37 and 52 m, respectively), high-span standard gas, prepared by adding appropriate amounts of pure $O_2$ or $N_2$ to industrially prepared $CO_2$ standard air, was introduced into the analyzers with the same flow rate and pressure as the
sample air and measured for 5 minutes, and then low-span standard gas was measured by the same procedure. The dilution effects on the $O_2$ mole fraction measured by the paramagnetic analyzer were corrected experimentally, not only for the changes in $CO_2$ of the sample air or standard gas measured by the NDIR, but also for the changes in Ar of the standard gas

measured by the mass spectrometer as $\delta(Ar/N_2)$. The analytical reproducibility of the $\delta(O_2/N_2)$ and $CO_2$ concentration achieved by the system was about 5 per meg and 0.06 $\mu mol\ mol^{-1}$, respectively, for 2-minute average values. Details of the

continuous measurement system used are given in Ishidoya et al. (2017).

It should be noted that we used the gravimetrically prepared air-based $CO_2$ standard gas system with uncertainties of $\pm0.13$ $\mu mol\ mol^{-1}$ on TU-10 scale (Nakazawa et al., 1991) to determine $CO_2$ concentration in this study. The highest concentration of the gravimetrically prepared standard gas was about 450 $\mu mol\ mol^{-1}$, while $CO_2$ concentrations of more than 600 $\mu mol$ $mol^{-1}$ were observed in this study. Therefore, we compared the NDIR-based $CO_2$ concentrations observed in this study with

those observed by using Cavity Ring-Down Spectroscopy (CRDS; G2401, Picarro) on NIES-09 scale (Machida et al., 2011) at the YYG site (our unpublished data). Although the highest $CO_2$ concentration of the gravimetrically prepared standard of the NIES-09 scale is similar to that of the TU-10 scale, a slope of 0.974 ppm $ppm^{-1}$ is derived from a least-squares regression line fitted to the relationship between the $CO_2$ concentrations observed by NDIR on the TU-10 scale and those by CRDS on the NIES-09 scale with a correlation coefficient ($r$) of 0.978. On the other hand, we obtained a slope of 1.002 per meg per

$meg^{-1}$ ($r = 0.999$) from the regression line fitted to the relationship between the $O_2$ concentrations of gravimetrically-prepared standard gases (Aoki et al., 2019) measured by the mass spectrometer on the AIST-scale and the gravimetric values of the standard gases covering a much wider range than the atmospheric variations in the $O_2$ concentration. Therefore, the uncertainty in OR due to the span-uncertainties of $O_2$ and $CO_2$ concentrations is expected to be within 3%.

In order to observe the $CO_2$ flux at the YYG site, the turbulence and the turbulent fluctuation of $CO_2$ were observed at 52 m

with a high time resolution of 10 Hz by using a sonic anemometer (WindMasterPro, Gill) and an open-path infra-red gas analyzer (LI-7500, LI-COR) since November 2012. The sensors were located at more than 5 times of mean building height (9 m), and then it was above the urban roughness sublayer. Turbulent flux of $CO_2$ was calculated by the eddy correlation method using EddyPro® (Licor) for every 30-minute period. Correlations were applied in the calculation for water-vapor density fluctuation (Webb et al., 1980) and mean vertical wind by using the double rotation algorithm (Wilczak et al., 2001).

The calculated flux was filtered for data quality based on the steady test and the integral turbulence characteristics in Aubinet et. al (2012). We used the flag 0 – 2 data in EddyPro® software based on Mauder and Foken (2006).

## 3. Results and discussion

### 3.1 Variations in the atmospheric $O_2$ and $CO_2$ concentrations

We show the 10-minute average values of the atmospheric $O_2$ and $CO_2$ concentrations observed at the height of 52 m at

YYG in Fig. 2. As seen in the figure, $O_2$ and $CO_2$ concentrations vary in opposite phase with each other on timescales ranging from several hours to seasonal cycle. In general, opposite phase variations of atmospheric $O_2$ and $CO_2$ are driven by fossil fuel combustion and terrestrial biospheric activities. In contrast, the atmospheric $O_2$ variation in $\mu mol\ mol^{-1}$ due to the air-sea exchange of $O_2$ is much larger than that of $CO_2$ on timescales shorter than 1 year (e.g. Goto et al., 2017; Hoshina et

al., 2018); this is because the equilibrium time for $O_2$ between the atmosphere and the surface ocean is much shorter than that for $CO_2$ due to the influence of the carbonate dissociation effect on the air-sea exchange of $CO_2$ (Keeling et al., 1993). Therefore, we attribute the opposite phase variations in $O_2$ and $CO_2$ observed in this study mainly to fossil fuel combustion and terrestrial biospheric activities. Figure 2 also shows that $\Delta O_2$, obtained by subtracting $O_2$ at 41 m from that at 52 m on the tower, varies in opposite phase with the corresponding $\Delta CO_2$. High $\Delta O_2$ values are more frequently observed in the winter than in the summer, and short-term (several hours to days) decreases in the $O_2$ concentration are intense in the winter.

To examine a relationship between the appearances of high $\Delta O_2$ and $O_2$ concentration decrease, detail variations in the $O_2$ and $CO_2$ concentrations, $\Delta O_2$ and $\Delta CO_2$ for the period December 16 – 23 and July 1 – 9, 2016 are shown in Fig. 3. As seen in the figure, increases in $\Delta O_2$ coincide with decreases in $O_2$ concentration in December, especially in the nighttime. Such coincidence is also seen in July, however, the increases in $\Delta O_2$ are much smaller than those in December. Therefore, it is highly likely that $O_2$ is consumed within the urban canopy at YYG, more so in the winter due to an increased usage of gas and/or liquid fuels for heating, and to a temperature inversion near the surface. The daily mean $CO_2$ flux from the urban area to the overlying atmosphere shown in Fig. 2 shows a seasonal cycle with a wintertime maximum, consistent with the enhancement of $O_2$ consumption in the urban canopy.

In this study, we focus on the short-term variations of $O_2$ and $CO_2$ for periods of several hours to days, to elucidate the $O_2$ and $CO_2$ exchange processes between the urban area and the atmosphere by examining two types of OR; one is $OR_{atm}$ calculated from a relationship between the $O_2$ and $CO_2$ concentration values observed at 52 or 37 m, and the other one is $OR_F$, for the $O_2$ and $CO_2$ fluxes between the urban area and the overlying atmosphere, calculated from a relationship between $\Delta O_2$ and $\Delta CO_2$. The relationships of the $O_2$ and $CO_2$ fluxes with $OR_F$ are based on the aerodynamic method of Yamamoto et al. (1999):

$$F_O = -K \frac{\Delta O_2}{\Delta z} \quad \text{(eq.2)}$$

$$F_C = -K \frac{\Delta CO_2}{\Delta z} \quad \text{(eq.3)}$$

$$OR_F = -\frac{F_O}{F_C} = -\frac{\Delta O_2}{\Delta CO_2} \quad \text{(eq.4).}$$

Here, $F_O$ ($F_C$) ($\mu mol\ m^{-2}s^{-1}$) represents the $O_2$ ($CO_2$) flux from the urban area to the overlaying atmosphere, K is the vertical diffusion coefficient, and $\Delta O_2 \Delta z^{-1}$ ($\Delta CO_2 \Delta z^{-1}$) is the vertical concentration gradient of $O_2$ ($CO_2$). The vertical diffusion is a sum of mass-independent eddy and mass-dependent molecular diffusion, however the effect of molecular diffusion on the observed variations of $O_2$ and $CO_2$ concentrations is generally negligible in the troposphere. It is significant in the stratosphere (e.g. Ishidoya et al., 2013a). Therefore, we used the same diffusion coefficient K for $O_2$ and $CO_2$ in eqs. (2) and (3), which enabled us to estimate $F_O$ by using the observed $\Delta O_2$, $\Delta CO_2$ and $F_C$ as in eq. (4). In general, $OR_{atm}$ reflects wider footprints of $O_2$ and $CO_2$ than $OR_F$ due to horizontal atmospheric transport (Schmid, 1994). We note that the definitions of $OR_F$ and $OR_{atm}$ are similar to those of $ER_F$ and $ER_{atm}$, respectively, reported by Ishidoya et al. (2013b, 2015).

In order to calculate $OR_{atm}$ for short-term variations, (1) we applied a best-fit curve consisting of the fundamental and its first harmonics (periods of 12 and 6 months) and a linear trend to the maxima (minima) values of $O_2$ ($CO_2$) observed at 52 m during the successive 1-week periods, and regarded the best-fit curve as its baseline variation, (2) then the baseline variation of $O_2$ ($CO_2$) concentration was subtracted from the respective $O_2$ ($CO_2$) concentrations observed at 52 m. Figure 4 shows the baseline variations and the variations in the $O_2$ and $CO_2$ concentrations observed at Minamitorishima (MNM; 24.28°N, 153.98°E), Japan (updated from Ishidoya et al., 2017). MNM is a small and isolated coral island located 1,850 km southeast of Tokyo, Japan, and the observation site was operated by the Japan Meteorological Agency (JMA) under the Global Atmosphere Watch program of the World Meteorological Organization (WMO/GAW). The baseline variations of $O_2$ and $CO_2$ at YYG show clear seasonal cycles with peak-to-peak amplitudes of 28 and 16 µmol mol$^{-1}$, respectively, with corresponding seasonal maximum and minimum appearing in mid August. The amplitude of the seasonal $O_2$ ($CO_2$) cycle and the appearance of seasonal maximum (minimum) were found to be larger and earlier, respectively, than those observed at MNM, while the annual average values of the baseline concentration variations of $O_2$ and $CO_2$ at YYG did not differ significantly from those at MNM. These characteristics of the seasonal cycles and the annual average values of the baseline variations at YYG and their comparison with those at MNM are generally consistent with those observed at similar latitude over the western Pacific region (Tohjima et al., 2005b). Therefore, in spite of the fact that the YYG site is located in a megacity, the baseline variations of $O_2$ and $CO_2$ concentrations are similar to those in the background air.

### 3.2 $O_2$:$CO_2$ exchange ratio between the urban area and the overlying atmosphere

Figure 5 (a) shows the relationship between all the $\Delta O_2$ and $\Delta CO_2$ values to obtain the average $OR_F$ throughout the observation period in this study. When errors in both species are non-negligible, a standard least-squares linear regression will give a biased and erroneous slope. Therefore, we apply an unweighted Deming regression analysis to the data (e.g. Linnet, 1993), assuming the ratio between the squared analytical standard deviations to be $0.06^2/(5 \times 0.2094)^2$ (ppm ppm$^{-1}$) to take into account the measurement uncertainties of $CO_2$ and $O_2$ concentrations. We regard the slope obtained by Deming regression to be $OR_F$, but we use a standard deviation obtained from a standard least-square regression to indicate the uncertainty of the slope. Jackknife method (Linnet, 1990) could be used to derive a standard error for Deming regression, however, by using a short dataset extracted from the observed data used in the present study, we confirmed that the standard deviations obtained from an ordinary regression are larger than the errors from the jackknife method. Therefore, using a standard deviation from ordinary regression is reasonable to ensure larger uncertainty for the $OR_F$. The average $OR_F$ value was calculated to be 1.620±0.004 (±1σ). This value falls within the range of the average OR values of 1.44 for liquid fuels and 1.95 for gas fuels, which suggests that the $O_2$ and $CO_2$ fluxes at YYG site were driven mainly by a consumption of liquid and gas fuels rather than terrestrial biospheric activities of which OR is about 1.1 (Severinghaus, 1995). The relationship between the $O_2$ and $CO_2$ concentration anomalies, calculated by subtracting the respective baseline variations shown in Fig. 4 from the observed $O_2$ and $CO_2$ concentrations, is also shown in Fig. 5 (b). By applying the Deming regression analysis to

the data, we obtained an average $OR_{atm}$ value of $1.541\pm0.002$ $(\pm1\sigma)$ throughout the observation period. The $OR_{atm}$ value also falls within the range of the average OR values for liquid fuels and gas fuels. However, the $OR_{atm}$ in this figure is not appropriate in representing the OR for the $O_2$ and $CO_2$ fluxes around the YYG site since it was determined by using the entire 18 months of collected observations that the site is influenced by various trajectories of air masses with much wider regional signature than the flux footprints. Therefore, we compare below the $OR_F$ and $OR_{atm}$ values by changing the aggregation periods to calculate the ORs and examine the validity of using $OR_F$ rather than $OR_{atm}$ to evaluate the relationship between the local $O_2$ and $CO_2$ fluxes.

Figure 6 shows examples of the $OR_F$ calculated by applying Deming regression fitted to $\Delta O_2$ and $\Delta CO_2$ values during the successive 12-hour periods observed in January, 2017 and July, 2016. The corresponding $OR_{atm}$ and wind direction observed for the periods are also shown in the figure. As seen in the figure, variabilities in the $OR_F$ and $OR_{atm}$ are larger in July than in December. The average $OR_F$, calculated using the OR values within a range of 0.5 to 2.5, were $1.65\pm0.20$ and $1.52\pm0.32$ in the winter (December to February) and summer (July to September), respectively. The corresponding average $OR_{atm}$ values were $1.61\pm0.15$ in the winter and $1.45\pm0.27$ in the summer. To examine the dependency of the OR on the wind direction, we also calculated $OR_F$ and $OR_{atm}$ for the periods when the prevailing wind directions were observed to be from $320° – 360°$ (NW) and $180° – 220°$ (SW) in the winter and summer, respectively. The number of measurements taken during the time of these prevailing winds constituted 30 % (winter) and 8 % (summer) of the total number of measurements. The calculated $OR_F$, $OR_{atm}$ and prevailing winds are shown by blue dots in Fig. 6. The average $OR_F$ ($OR_{atm}$) values, calculated using the OR values within a range of 0.5 to 2.5, were $1.65\pm0.25$ ($1.58\pm0.19$) in the winter and $1.58\pm0.40$ ($1.42\pm0.33$) in the summer, respectively. Therefore, the average $OR_F$ and $OR_{atm}$ calculated using all the values obtained from the 12-hour aggregation periods did not differ significantly from those that were calculated using only the data that were associated with the above-mentioned prevailing wind directions. The average $OR_F$ seems to be slightly higher than $OR_{atm}$, however, their uncertainties are too large to discuss the significance of the slight difference. Taking these facts into consideration, we use all the $O_2$ and $CO_2$ concentration data without filtering by the wind direction, to increase the number of data points for calculating $OR_F$ and $OR_{atm}$; this is consistent with the purpose of this study to derive representative OR values at the YYG site in order to validate the $CO_2$ emission inventory (Hirano et al., 2015). For analyses of specific events, we have reported analytical results of $OR_{atm}$ and simultaneously-measured $PM_{2.5}$ aerosol composition for a week-long pollution event at the YYG site (Kaneyasu et a., 2020).

To examine the seasonal difference between the $OR_F$ and $OR_{atm}$ values, we show the $OR_F$ values calculated by applying regression lines to 1 day and 1 week successive $\Delta O_2$ and $\Delta CO_2$ values in Fig. 7. The corresponding $OR_{atm}$ values, obtained by applying Deming regression fitted to successive $O_2$ and $CO_2$ concentrations anomalies in Fig. 5 (b), are also shown. Since there is no statistically significant difference between the two (based on the uncertainties shown in the figure ($\pm1\sigma$)), we focus our discussion on the OR values obtained from the 1 week successive data. Clear seasonal cycles with wintertime maxima are found both in the $OR_F$ and $OR_{atm}$ values at YYG. Larger $OR_{atm}$ values in the winter than in the summer in urban areas have been reported by some past studies (e.g. van der Laan et al., 2014; Ishidoya and Murayama, 2014; Goto et al.,

2013), and generally interpreted as a result of the wintertime increase and decrease of fossil fuel combustion and terrestrial biospheric activities, respectively. Biospheric activities included in the summertime and wintertime flux footprints at YYG were 9 and 2%, respectively (Hirano et al., 2015), and there was no significant solid fuel consumption, such as coal-fired power generation plant of which OR is expected to be 1.17 (Keeling, 1988), detected in the footprints. At YYG, the effect of emissions from coal combustion is evaluated simultaneously by the use of aerosol composition monitored every 4 hours (Kaneyasu et al., 2020). From these measurements, emission contribution from coal combustion can be detected under a limited meteorological condition, such as stagnant condition under weak south-southwesterly wind. This condition occurred only several times a year, mostly from spring to fall. Therefore, the wintertime $OR_F$ was determined mainly by gas and liquid fuels consumption around the YYG site, given that little vegetation and weak terrestrial biospheric activities occurred in the wintertime. If we assume the wintertime $OR_F$ is determined only by gas and liquid fuels consumption, with OR values of 1.95 and 1.44, respectively, then 45% of the $CO_2$ flux during the December to February (DJF) period was driven by gas fuel consumption, with the rest attributed to liquid fuel consumption. It should be noted that the contributions of gas and liquid fuels are expected to be under- and overestimated since we have ignored the contribution from human respiration with OR values in the range of 1.0 to 1.4. The respiration quotients (the reciprocal of OR) for carbohydrates, lipid and protein are known to be about 1.0, 0.7 and 0.8, respectively. We also conducted detail analyses to separate out the contributions from the consumption of gas and liquid fuels and human respiration by using the observed $CO_2$ flux and $OR_F$, and comparing the results with the $CO_2$ emission inventory in 3-3.

Figure 7 also shows that the $OR_F$ values were systematically larger than $OR_{atm}$ throughout the year, except for October 2016 and July 2017. The average $OR_F$ and $OR_{atm}$ during DJF were 1.67±0.03 and 1.63±0.02, respectively, both of which agree with the OR value of 1.65 calculated using the statistical data of fossil fuel consumption in Tokyo reported by the Agency of Natural Resources and Energy (http://www.enecho.meti.go.jp/en/), assuming OR value of 1.95, 1.44 and 1.17 for gas, liquid and solid fuels consumption, respectively (hereafter referred to as "$OR_{ff}$"). By using the same procedure as above, the average $OR_{ff}$ was calculated to be 1.52±0.1 for the Kanto area of about 17,000 km$^2$ that includes Tokyo. Therefore, it is suggested not only $OR_F$ but also $OR_{atm}$ at YYG mainly reflected an influence of the fossil fuel consumption in Tokyo rather than that in the wider Kanto area in the wintertime. Both the $OR_F$ and $OR_{atm}$ values in the summer were lower than $OR_{ff}$ in Tokyo (1.65), but $OR_{atm}$ was also found to be lower than $OR_{ff}$ for the Kanto area (1.52). These lower $OR_F$ and $OR_{atm}$ values, compared to those of the $OR_{ff}$ suggest that the ratio of fossil fuel combustion to terrestrial biospheric activities and human respiration is lower in the summer than that in the winter. The slightly lower $OR_{atm}$ than $OR_F$ at YYG throughout the year is probably due to the higher contribution of the air mass from Kanto area to $OR_{atm}$ than $OR_F$, since the Kanto area as a whole has lower $OR_{ff}$ than for Tokyo; in addition, the south Kanto area (including Tokyo) has a larger vegetation coverage of about 50% than that in the area around YYG site. From the comparison results of the $OR_F$ with $OR_{atm}$ in Fig. 5 − 7, it is suggested that the $OR_{atm}$ reflects wider footprints of $O_2$ and $CO_2$ than $OR_F$ for the aggregation periods at least longer than 12 hours to calculate the $OR_{atm}$. Therefore, to use $OR_F$ rather than $OR_{atm}$ is more appropriate to validate inventory-based $CO_2$ emissions from gas, liquid and solid fuels in the flux footprint.

### 3.3 Consumption of gas and liquid fuels estimated from the observed $CO_2$ flux and $O_2:CO_2$ exchange ratio for net turbulent flux

In this section, we derive average diurnal cycles of $OR_F$, $CO_2$ and $O_2$ flux and estimate the $CO_2$ fluxes from gas and liquid fuels consumption separately. Figure 8 shows the average diurnal cycles of $\Delta O_2$ and $\Delta CO_2$ for each season. To derive the average diurnal cycles, the observed $\Delta O_2$ and $\Delta CO_2$ values of each day in a season were overlaid on top of the values of other days, added up and divided by the number of days in the season. The error bars shown in Fig. 8 indicate $\pm 1$ standard error ($\sigma/\sqrt{n}$). The $\Delta O_2$ and $\Delta CO_2$ values vary systematically in opposite phase and take positive and negative values respectively, indicating transport of $O_2$ uptake and $CO_2$ emission signals from the urban area to the overlying atmosphere throughout the year. Daily maxima of $\Delta O_2$ shown in Fig. 8 are higher in the winter than in the summer and occur in the nighttime. These characteristics would be attributable to an enhancement of the anthropogenic $O_2$ consumption in the winter, while the nighttime decrease of $O_2$ concentration would be due to the $O_2$ consumption near the surface and temperature inversion near the surface. It must be noted that the $\Delta CO_2$ values in the daytime are nearly zero, while the $\Delta O_2$ values are not. The intercepts of the regression lines fitted to the relationship between $\Delta O_2$ and $\Delta CO_2$ in Fig. 8 are 0.27, 0.41, 0.45 and 0.44 $\mu mol \ mol^{-1}$ in DJF, MAM, JJA and SON, respectively. Unfortunately, we did not fix the cause(s) of such biases yet, although it may be related, to some extent, to natural exchange processes between the urban area and the overlying atmosphere. Therefore, because of these issues, the use of $OR_F$, calculated by applying a Deming regression fitted to 2-hour period values of $\Delta O_2$ and $\Delta CO_2$ of the climatological diurnal cycle (the number of data included in each 2-hour periods were 400 – 800, depending on the season), to determine the relationship between the $O_2$ and $CO_2$ fluxes is preferable. The $OR_F$ values plotted in Fig. 8 show diurnal cycles with daytime minima in DJF, MAM and SON while no clear cycle is found in JJA. From 10:00 – 16:00 local time, the $OR_F$ values are in the range of 1.44 – 1.59 for all seasons. On the other hand, the $OR_F$ values from 18:00 – 9:00 local time are more variable, in the range of 1.39 – 1.74, and are clearly larger in the winter than in the summer.

The observed $CO_2$ flux and the estimated $O_2$ flux for each season are shown in Fig. 8. The $CO_2$ flux shows clear diurnal cycles with two peaks for all seasons, one in the morning and the other in the evening. The shape of the diurnal $CO_2$ flux cycle, with larger flux in the winter than in the summer, was also found in our previous study at YYG for the period 2012-2013 (Hirano et al., 2015). On the other hand, the $O_2$ flux shows similar diurnal cycles but in opposite phase with the $CO_2$ flux. The daily mean $CO_2$ fluxes were $15.6 \pm 0.2$, $11.2 \pm 0.1$, $9.3 \pm 0.1$ and $11.5 \pm 0.1$ $\mu mol \ m^{-2}s^{-1}$ in DJF, MAM, JJA and SON, respectively, while the respective daily mean $O_2$ fluxes were $-25.4 \pm 0.3$, $-17.8 \pm 0.2$, $-14.1 \pm 0.2$ and $-17.7 \pm 0.2$ $\mu mol \ m^{-2}s^{-1}$. The annual average daily mean $O_2$ flux was $-16.3 \ \mu mol \ m^{-2}s^{-1}$. Steinbach et al. (2011) reported a global dataset of $CO_2$ emissions and $O_2$ uptake associated with fossil fuel combustion using the EDGAR inventory with country level information on OR, based on the fossil fuel consumption data from the UN energy statistics database. The $O_2$ uptake around Tokyo for the year 2006 has been shown to be about $e^{16} - e^{17} \ kgO_2 \ km^{-2} \ year^{-1}$ (Fig. 2 in Steinbach et al (2011)), which corresponds to $-9 - -24 \ \mu mol \ m^{-2}s^{-1}$ of $O_2$ flux and is consistent with those observed in this study. In this regard, the atmospheric $O_2$

concentration decreased secularly due mainly to fossil fuel combustion at a rate of change of about -4 μmol yr$^{-1}$ (e.g. Keeling and Manning 2014), corresponding to -0.04 μmol m$^{-2}$s$^{-1}$ of $O_2$ flux, assuming 5.1 x 10$^{14}$ m$^2$ for the surface area of the earth, 5.124 x 10$^{21}$ g for the total mass of dry air (Trenberth, 1981) and 28.97 g mol$^{-1}$ for the mean molecular weight of dry air. Therefore, the consumption rate of atmospheric $O_2$ in an urban area of Tokyo is several hundred times larger than the global mean surface consumption rate.

The $CO_2$ emission inventory was developed based on Hirano et al. (2015) with some modifications. We added human respiration based on the hourly population data (Regional Economy Society Analyzing System, https://resas.go.jp/). Respiration amount per person was referred from Moriwaki and Kanda (2004). We also added $CO_2$ emission due to gas consumption by restaurants to the Hirano et al. (2015) inventory which only accounted for household emission. Monthly gas consumption in restaurants was acquired from the statistical data published by the local government (http://www.toukei.metro.tokyo.jp/tnenkan/2015/tn15q3i006.htm). Diurnal variation in the gas consumption by the restaurants was obtained from Takahashi et al. (2006) and Takada et al. (2007). We also modified the household gas consumption using the study by Etsuki (2010). As for the traffic, we used a traffic load data (http://www.jartic.or.jp/) which recorded the number of vehicles on the road every hour every day, whereas Hirano et al. (2015) used traffic data for a single day in 2010.

The $OR_F$ is determined as a ratio of net turbulent fluxes of $O_2$ and $CO_2$ from mixed consumption of gas, liquid and solid fuels and terrestrial biospheric activities and human respiration. In this study, the total net turbulent $CO_2$ flux from the urban area to the overlying atmosphere is calculated using the eddy correlation method. The $CO_2$ emission inventories from gas consumption, traffic and human respiration have also been updated from the original data published by Hirano et al. (2015). We can then proceed to separate out the $CO_2$ flux from gas and liquid fuels consumption by using eq. (4), followed by eqs. (5)-(6):

$$F_O = -(OR_G \times F_G + OR_L \times F_L + OR_R \times F_R) \qquad \text{(eq.5)}$$

$$F_C = F_G + F_L + F_R \qquad \text{(eq.6)}$$

where $F_G$, $F_L$ and $F_R$ (μmol m$^{-2}$s$^{-1}$) represent the $CO_2$ fluxes from gas and liquid fuels consumption and human respiration from the urban area to the overlaying atmosphere, and $OR_G$, $OR_L$ and $OR_R$ are the OR values for gas and liquid fuels consumption and human respiration, respectively. We use 1.95, 1.44 and 1.2 for $OR_G$, $OR_L$ and $OR_R$, respectively. For this analysis, it is assumed that the contributions from solid fuels consumption and terrestrial biospheric activities are negligible, given the fact that in the flux footprint area, significant solid fuel consumption is absent and the vegetated area is relatively small. We also assume $OR_R$ value of 1.2 as an intermediate value of the reciprocal of respiration quotients for carbohydrates, lipid and protein. We use the $F_c$ observed by the eddy correlation method and the $F_R$ obtained from the $CO_2$ emission inventory to estimate $F_G$ and $F_L$.

Figure 9 shows average diurnal cycles of the observed total $CO_2$ flux, and the $CO_2$ flux from gas and liquid fuels consumption estimated from eqs. (4)-(6) for each season. The average diurnal cycles of the inventory-based total, gas, traffic and human respiration $CO_2$ fluxes are also shown in the figure. As seen in Fig. 9, similar diurnal cycles with two peaks are

found both in the observed and inventory-based total $CO_2$ fluxes for all seasons. Two peaks of the diurnal cycles are also found in the diurnal cycles of the estimated and inventory-based $CO_2$ fluxes from gas consumption, however, the evening peaks of the inventory-based flux in MAM, JJA and SON are clearly larger than the estimated values. It is also seen from the figure that the diurnal cycles of inventory-based traffic $CO_2$ flux do not change significantly throughout the year, while those of the estimated $CO_2$ flux from liquid fuels consumption shows large variabilities especially in the morning. Such variability may be caused by the smaller $\Delta O_2$ and $\Delta CO_2$ values observed during the daytime, compared to those in the nighttime, as well as due to a rapid change in the atmospheric stability after the daybreak. The actual diurnal cycles of liquid fuels consumption do not seem to change significantly throughout the year, considering the results of the inventory-based traffic $CO_2$ flux. We therefore regard the standard deviations of the seasonal diurnal cycles of the estimated $CO_2$ flux from liquid fuels consumption from the annual average diurnal cycle to be the uncertainties for the annual average cycle.

Figure 10 shows the same diurnal cycles of the observed, estimated, and inventory-based $CO_2$ fluxes as in Fig. 9 but for the annual average cycle. The observed total $CO_2$ flux is found to be significantly smaller than the inventory-based flux in the evening. Similar discrepancy was also seen in our previous study (Hirano et al., 2015). The main cause for this discrepancy in the evening is likely due to the much larger inventory-based $CO_2$ flux from gas consumption than the estimated flux. The estimated $CO_2$ flux from liquid fuels consumption is somewhat larger than the inventory-based traffic $CO_2$ flux in the evening, thus contributing to the above-mentioned discrepancy to some extent. Although the uncertainty in the estimated $CO_2$ flux is large in the morning, the observed peak of the estimated $CO_2$ flux from gas fuels consumption early in the morning and the gradual increase of the estimated $CO_2$ flux from liquid fuels consumption over the same time period can be distinguishable. Such temporal variations of the estimated $CO_2$ flux are reasonable since gas fuels consumption for domestic heating and cooking should increase early in the morning and liquid fuels consumption from the traffic should increase during the morning commute. Consequently, it is confirmed that the simultaneous observations of the $OR_F$ and $CO_2$ flux are useful in validating the $CO_2$ emission inventories developed based on statistical data. However, as shown in Figs. 8 − 10, a large number of $\Delta O_2$ and $\Delta CO_2$ measurement data is needed to derive reliable $OR_F$ based on an aerodynamic method. If we measure $O_2$ concentration with high time-resolution to determine net turbulent $O_2$ flux by an eddy correlation method, then it will be possible to derive high time-resolution $OR_F$ as a ratio of the observed $O_2$ to $CO_2$ fluxes. Such an innovative technique will enhance the value of the $OR_F$ observations significantly for an evaluation of the urban $CO_2$ emissions.

## 4. Conclusions

Continuous simultaneous observations of atmospheric $O_2$ and $CO_2$ and $CO_2$ flux have been carried out at the YYG site, Toyo, Japan since March 2016. Sample air was taken from air intakes set at heights of 52 m and 37 m of the YYG tower, allowing us to apply an aerodynamic method by using the vertical gradients of the $O_2$ and $CO_2$ concentration measurements. We compared $OR_F$ obtained from the aerodynamic method with $OR_{atm}$, representing OR of the overlying atmosphere above the

urban canopy. We found clear seasonal variations with wintertime maxima for both $OR_F$ and $OR_{atm}$, as well as slightly
higher $OR_F$ than $OR_{atm}$ throughout the year. The annual mean $OR_F$ and $OR_{atm}$ were observed to be 1.62 and 1.54, respectively, falling within the range of the respective average OR values of 1.44 and 1.95 of liquid and gas fuels. The slightly lower $OR_{atm}$ than $OR_F$ throughout the year was probably due to an influence of the air mass from the wider Kanto area to $OR_{atm}$ at YYG since the OR value of 1.1 for the terrestrial biospheric activities is lower than those for liquid and gas fuels consumption; in addition, the influence of the vegetation included in the flux footprints at YYG was much smaller than that in the surrounding Kanto area. Therefore, we prefer to use $OR_F$ rather than $OR_{atm}$ to validate the inventory-based $CO_2$ emissions from gas, liquid and solid fuels in the YYG flux footprint region.

Seasonal variations were seen in the average diurnal $OR_F$ cycles, showing daytime minima in DJF, MAM and SON, while no clear diurnal cycle was distinguishable in JJA. The daily mean $O_2$ flux at YYG, calculated from the $OR_F$ and $CO_2$ flux, was about -25 and -14 $\mu$mol m$^{-2}$s$^{-1}$ in the winter and the summer, respectively, which means the consumption rate of atmospheric $O_2$ in an urban area of Tokyo is several hundred times larger than the global mean surface consumption rate. We estimated the average diurnal cycles of $CO_2$ flux from the consumption of gas and liquid fuels for each season, based on the average diurnal cycles of $OR_F$ and $CO_2$ flux, and the $CO_2$ emission inventory of human respiration around the YYG site. Discrepancy between the estimated and inventory-based $CO_2$ fluxes from gas fuels consumption was found to be the main cause of the significantly smaller evening peak of the observed total $CO_2$ flux than that of the inventory-based total flux. Along with the peak in the estimated $CO_2$ flux from the gas fuels consumption, the gradual increase in the estimated $CO_2$ flux from the liquid fuels consumption found in the morning is consistent with the fact that the gas fuels consumption for domestic heating and cooking, and liquid fuels consumption from traffic during commuting occur in the morning. Therefore, we can use simultaneous observations of $OR_F$ and $CO_2$ flux as a powerful tool to validate $CO_2$ emission inventories obtained from statistical data.

*Data availability.*

The data at YYG site presented in this study can be accessed by contacting the corresponding author.

*Author contributions.*

SI designed the study and drafted the manuscript. Measurements of $O_2$ concentrations, $CO_2$ concentrations, and $CO_2$ flux were conducted by SI, SI and YT, and HS, respectively. HS prepared $CO_2$ emission inventory data. NA prepared standard gas for the $O_2$ measurements. SI and KT conducted $O_2$ observations at MNM. HS, NK and HK examined the results and provided feedback on the manuscript. All the authors approved the final manuscript.

*Competing interests.*

The authors declare that they have no conflict of interest.

**Acknowledgements.**

We thank Prof. T. Nakajima at Tokai University, Dr. Shohei Murayama and JANS Co. Ltd. for supporting the observation.
This study was partly supported by the JSPS KAKENHI Grant Number 24241008, 15H02814 and 18K01129, and the
Environment                                                    Research                                                    and
Technology Development Fund (1-1909) and the Global Environment Research Coordination System from the Ministry of
the Environment, Japan.

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

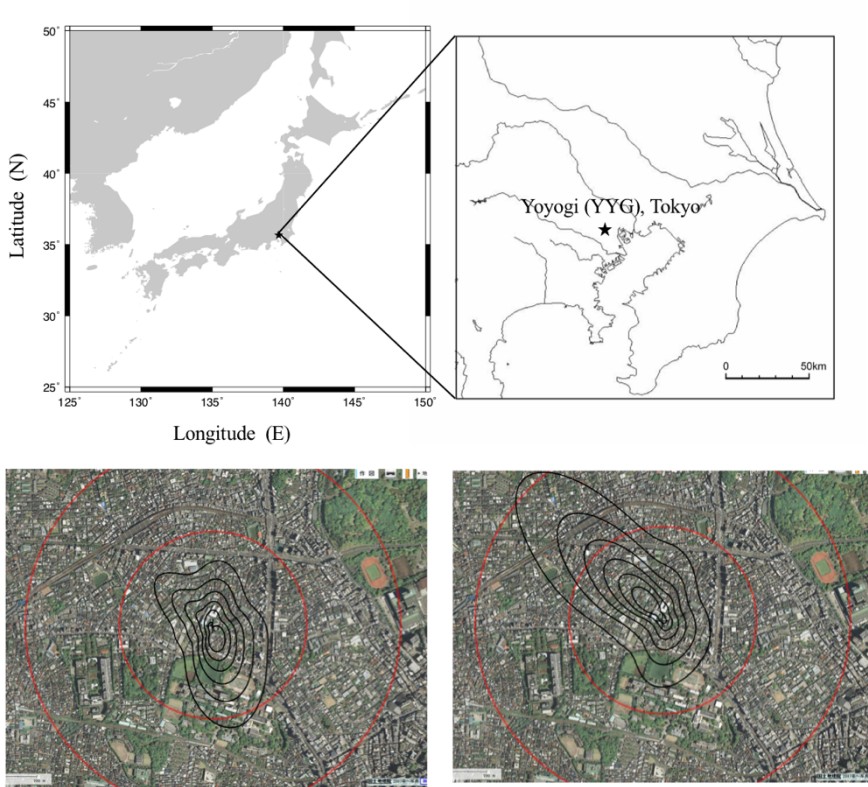

**Figure 1: Upper panel: Location of the Yoyogi site (35.66°N, 139.68°E, YYG), Tokyo, Japan. Lower panel: Aerial photo from the Geospatial Information Authority of Japan around the study area at YYG. Ensemble-mean flux footprints in the summer (left) and the winter (right) are also shown by black circles. The contour lines indicate contribution in measured flux (60, 50, 40, 30, 20 and 10% from outside to inside). Inside and outside the red circles indicate the distance of 500 m and 1000 m, respectively, from a roof-top tower of Tokai University where the observations of $O_2$ and $CO_2$ concentrations and $CO_2$ flux were carried out.**


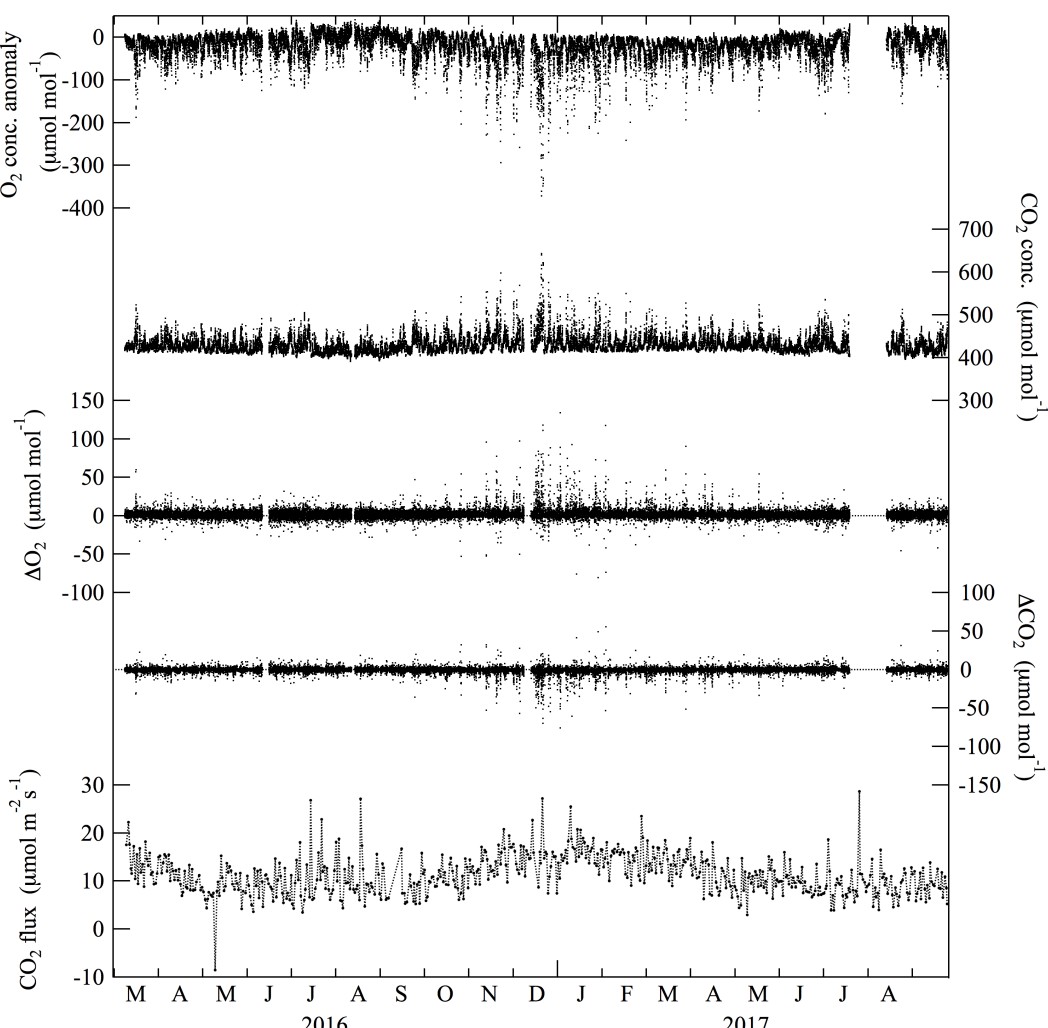

Figure 2: Variations in $O_2$ and $CO_2$ concentrations observed at the tower height of 52 m at Yoyogi, Tokyo, Japan for the period March 2016 – September 2017. The $O_2$ concentrations are expressed as deviations from the value observed at 9:58 on March 9, 2016. $\Delta O_2$, representing the differences calculated by subtracting the observed $O_2$ concentrations at 37 m from that at 52 m, are also shown. $\Delta CO_2$ are the same as $\Delta O_2$ but for $CO_2$ concentration. Daily mean $CO_2$ fluxes observed using the eddy correlation method are also shown, and the flux takes on positive value when the urban area emits $CO_2$ to the overlying atmosphere.



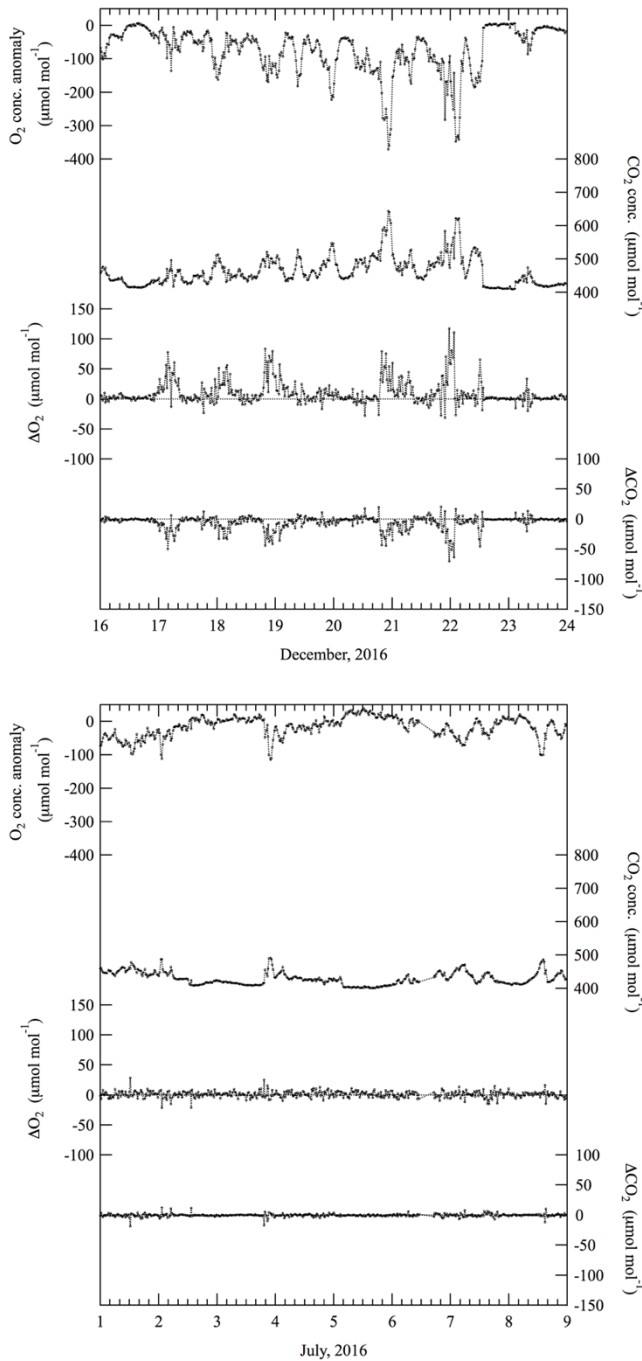

Figure 3: Same as in Fig. 2 but for O₂ and CO₂ concentrations, ΔO₂ and ΔCO₂ for the period December 16 – 23 and July 1 – 9, 2016.

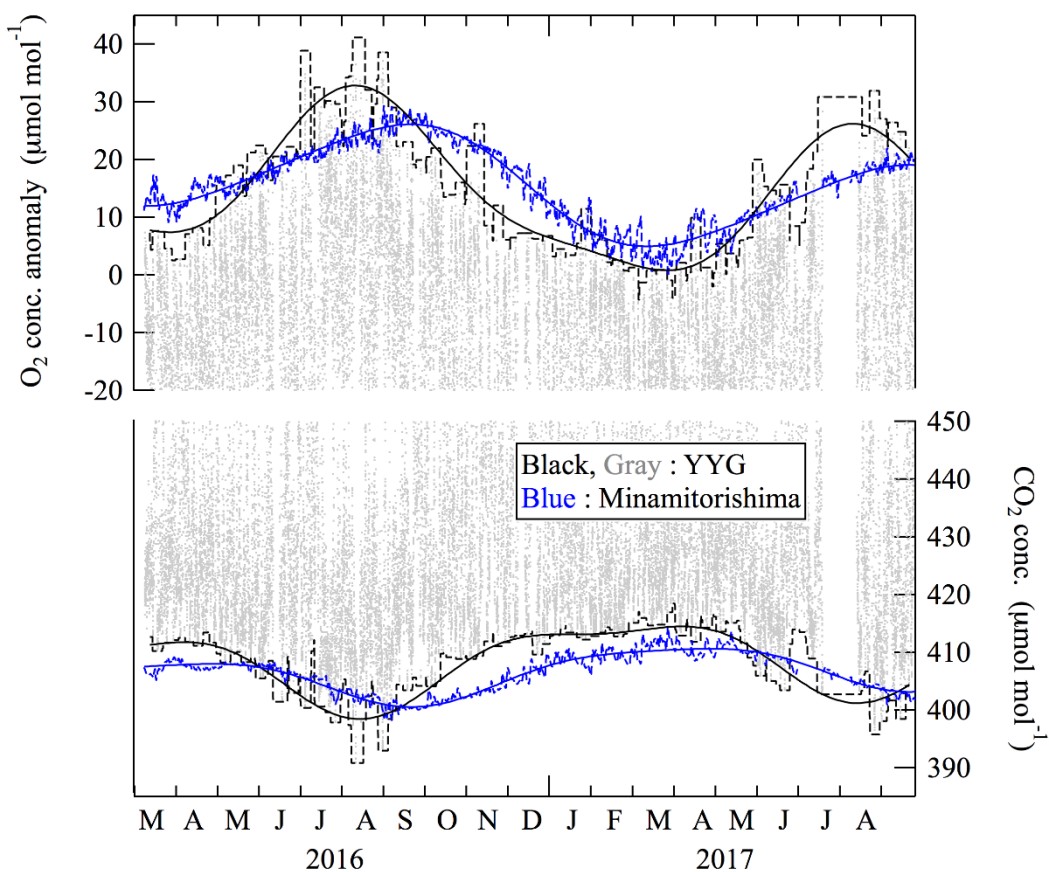

**Figure 4: Baseline variations of $O_2$ and $CO_2$ concentrations at the tower height of 52 m at Yoyogi, Tokyo, Japan, represented by their best-fit curves (black solid lines) to the respective maxima and minima values during the successive 1 week periods (black dashed lines). Variations of 24 hours-averaged $O_2$ and $CO_2$ concentrations at Minamitorishima, Japan (blue dashed line) and their best-fit curves (blue solid lines) are also shown (updated from Ishidoya et al., 2017).**


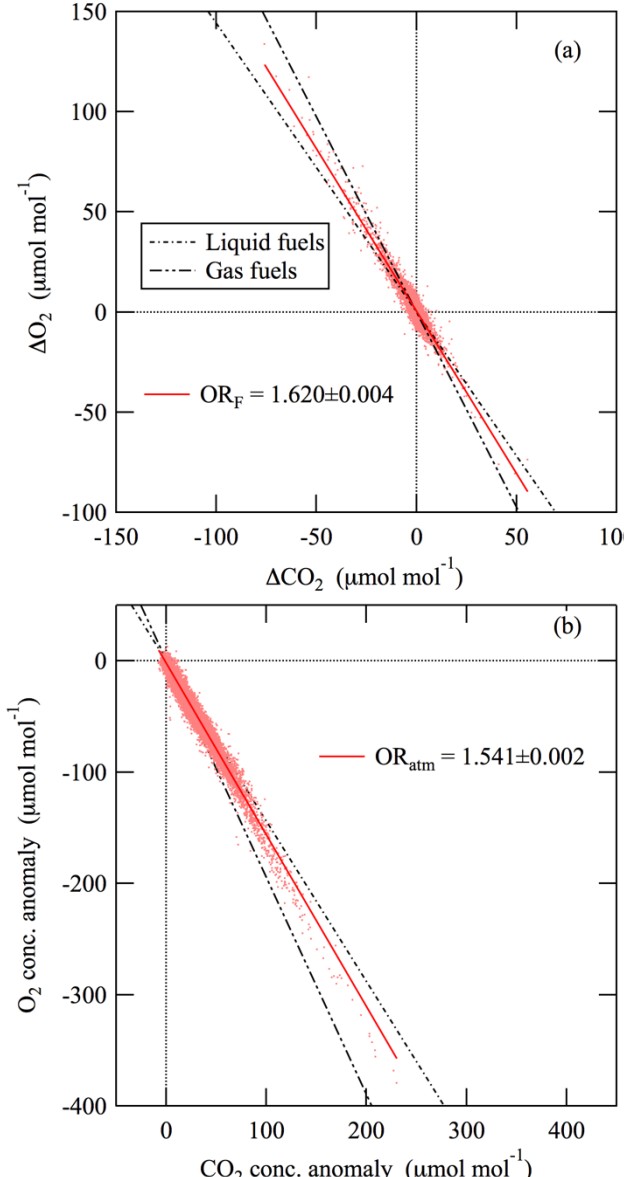

Figure 5: (a) Relationship between the $\Delta O_2$ and $\Delta CO_2$ shown in Fig. 2. Average $OR_F$ (see text) for the observation period, derived from the Deming regression fitted to the data is also shown. (b) Same as in (a) but for the deviations of $O_2$ and $CO_2$ concentrations from their baseline variations shown in Fig. 3 and the average $OR_{atm}$ (see text). OR values expected from the consumptions of gas and liquid fuels are also shown.

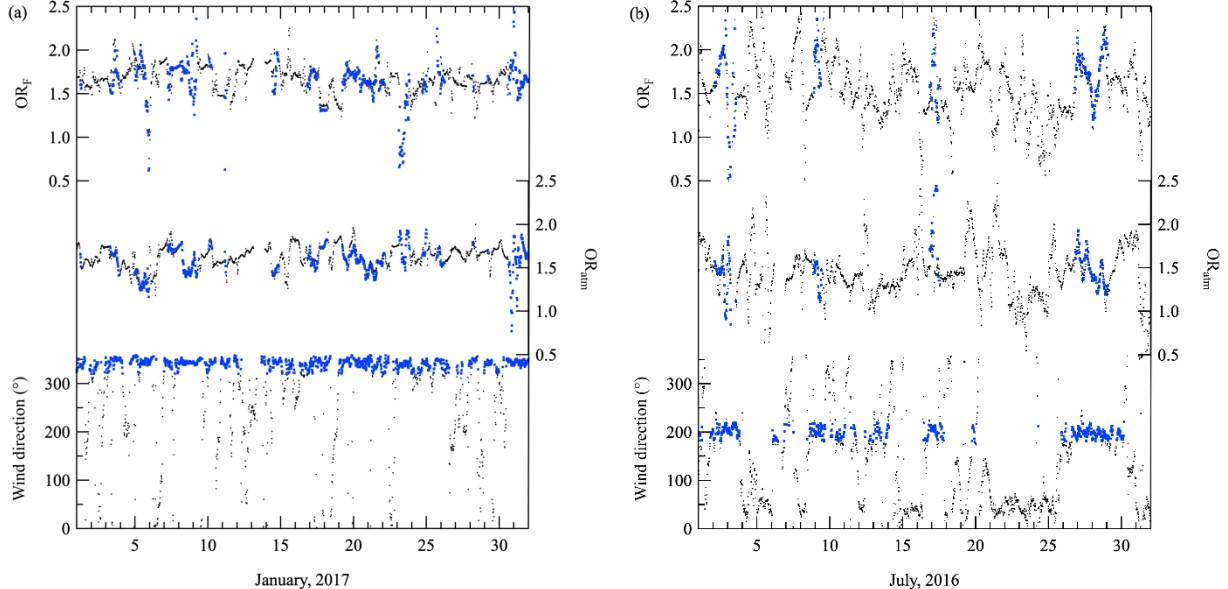

**Figure 6: (a) $OR_F$ (black dots, top) calculated by applying Deming regression fitted to $\Delta O_2$ and $\Delta CO_2$ values during the successive 12-hour periods observed in January, 2017. The corresponding $OR_{atm}$ (black dots, middle) obtained from the deviations of $O_2$ and $CO_2$ concentrations from their baseline variations shown in Fig. 4, and the wind directions (black dots, bottom) are also shown. Angles of 90°, 180°, 270° and 360° for the wind direction denote winds from east, south, west and north, respectively. The $OR_F$ and $OR_{atm}$ obtained from the data observed during the period with the prevailing wind direction (blue dots, bottom) are also shown by blue dots. (b) Same as in (a) but for July, 2016.**

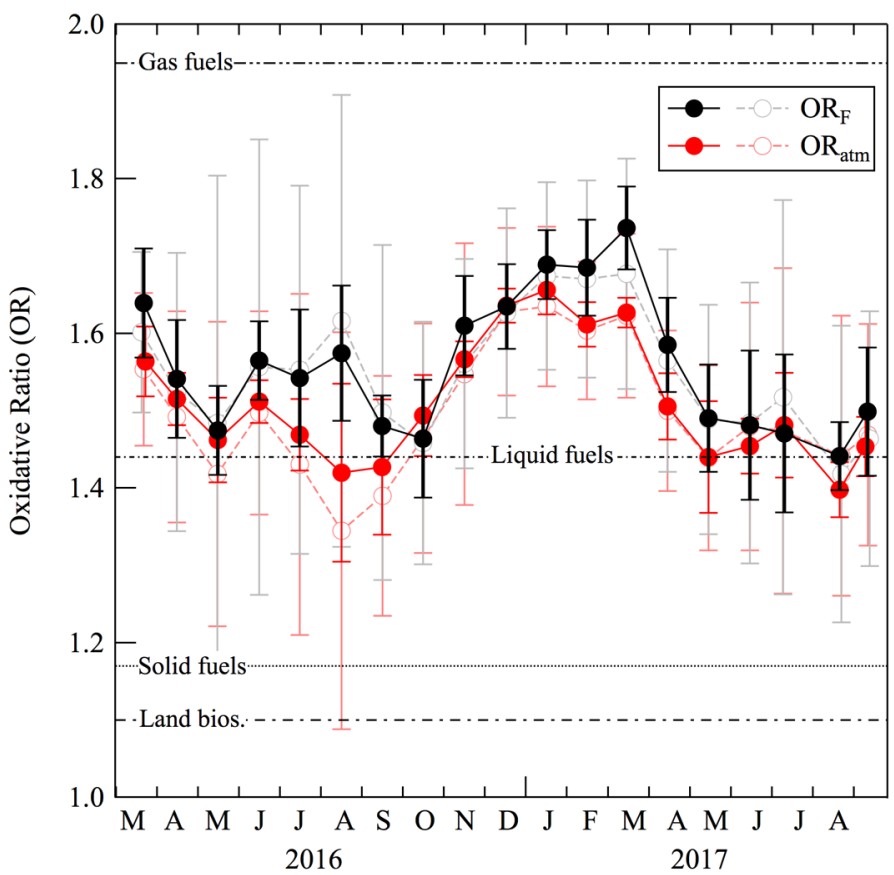


**Figure 7: OR$_F$ calculated by applying Deming regression fitted to 1 day (gray open circles) and 1 week (black closed circles) successive $\Delta O_2$ and $\Delta CO_2$ values. Also plotted are OR$_{atm}$ calculated by applying Deming regression fitted to 1 day (light red open circles) and 1 week (dark red closed circles) successive $O_2$ and $CO_2$ deviations from their baseline variations shown in Fig. 3. OR values expected from the consumptions of gas, liquid and solid fuels and land biospheric activities are also shown.**


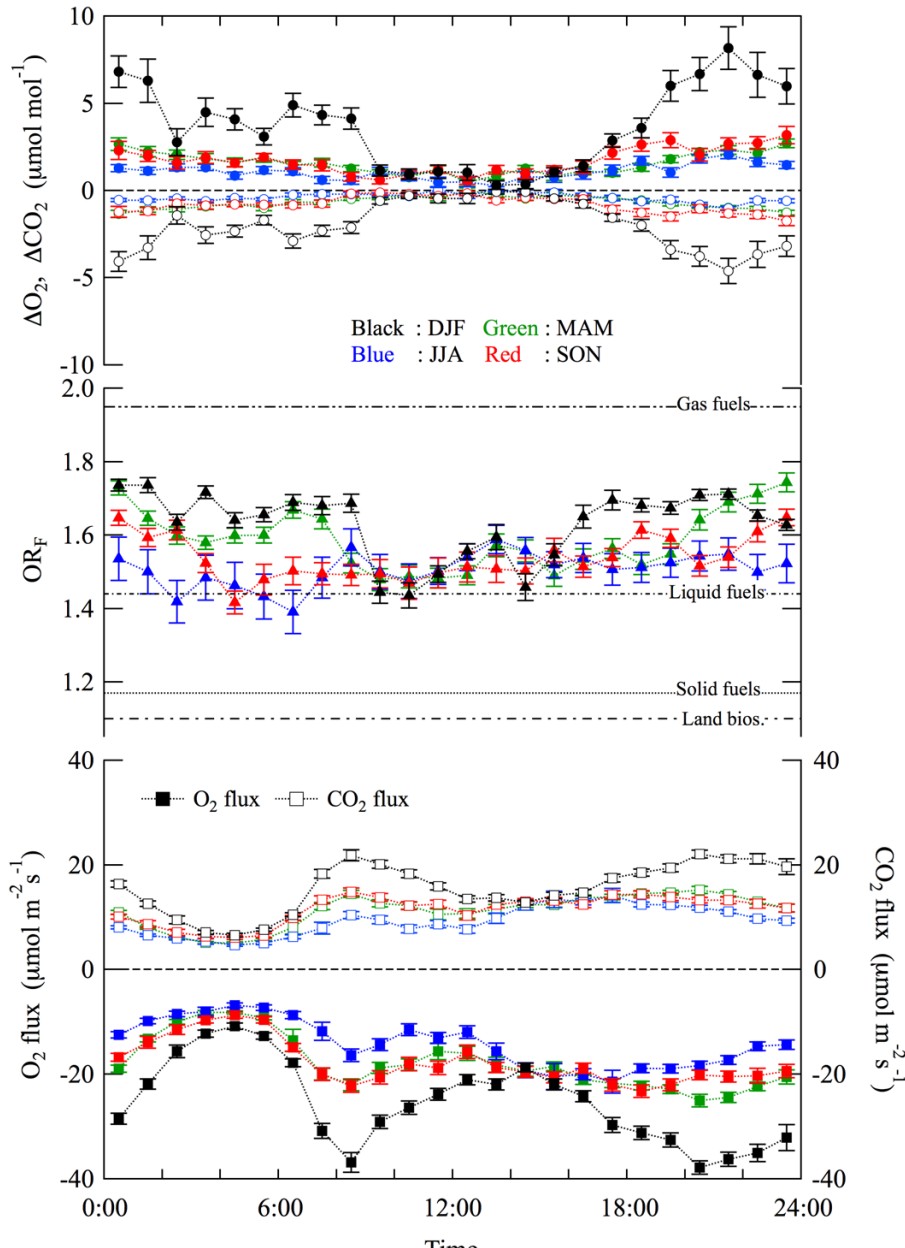

**Figure 8: Plots of average diurnal cycles of $\Delta O_2$ (filled circles) and $\Delta CO_2$ (open circles) for each season: December to February (back), March to May (green), June to August (blue) and September to November (red). Average diurnal cycles of $OR_F$, calculated by applying Deming regression fitted to the 2-hour period values of $\Delta O_2$ and $\Delta CO_2$, are also plotted seasonally (see text). Average diurnal cycles of the $CO_2$ flux observed using the eddy correlation method, and those of the $O_2$ flux calculated from the $CO_2$ flux and $OR_F$ values are also plotted seasonally. Error bars indicate ±1 standard error.**


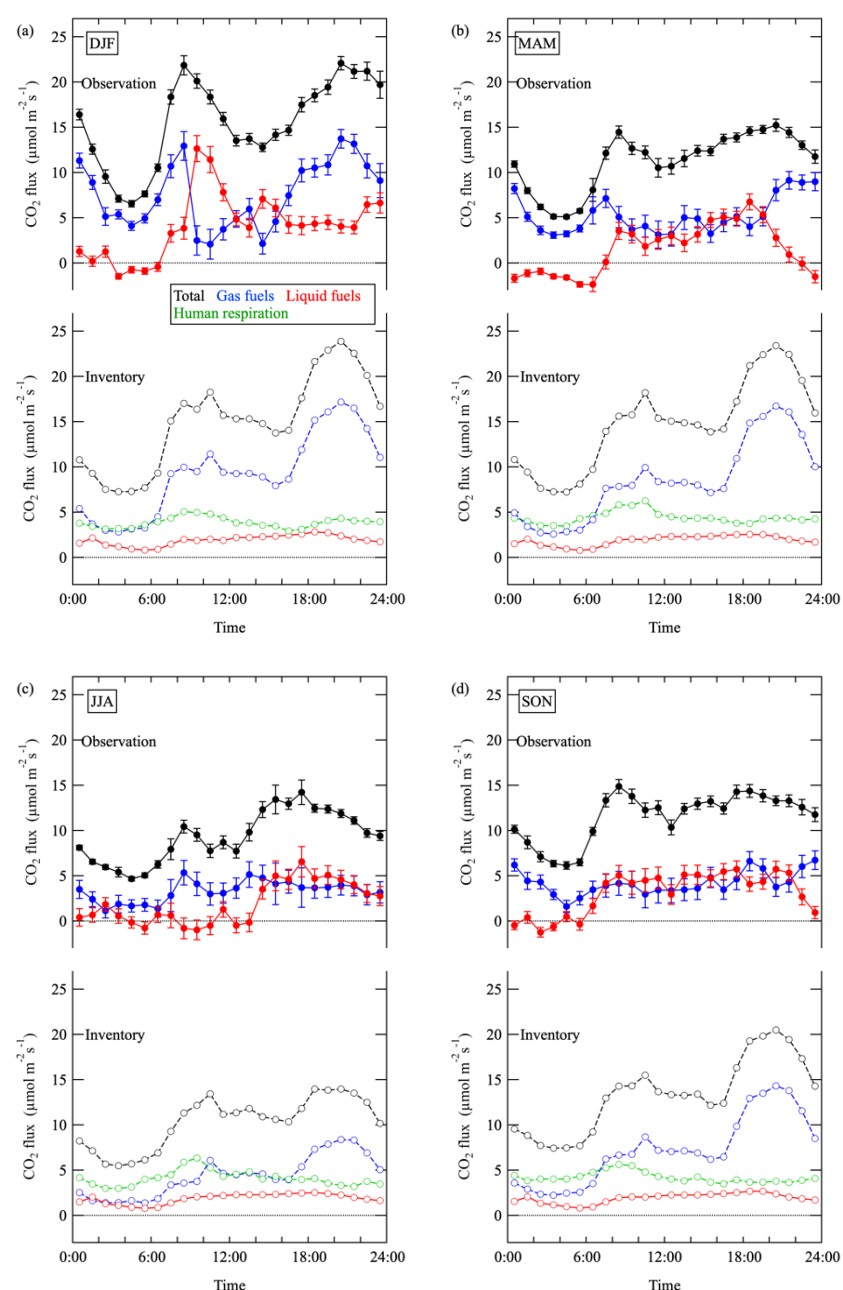


**Figure 9: Average diurnal cycles of the total CO₂ flux observed using the eddy correlation method (black filled circles), the estimated CO₂ flux from gas (blue filled circles) and liquid fuels (red filled circles) consumption by using the total CO₂ flux and OR_F for each season: December to February (a), March to May (b), June to August (c) and September to November (d). Average diurnal cycles of the CO₂ emission inventory of gas consumption (blue open circles), traffic (red open circles), human respiration**

**(green open circles) and their total (black open circles) around YYG are also shown for each season. See text in detail.**

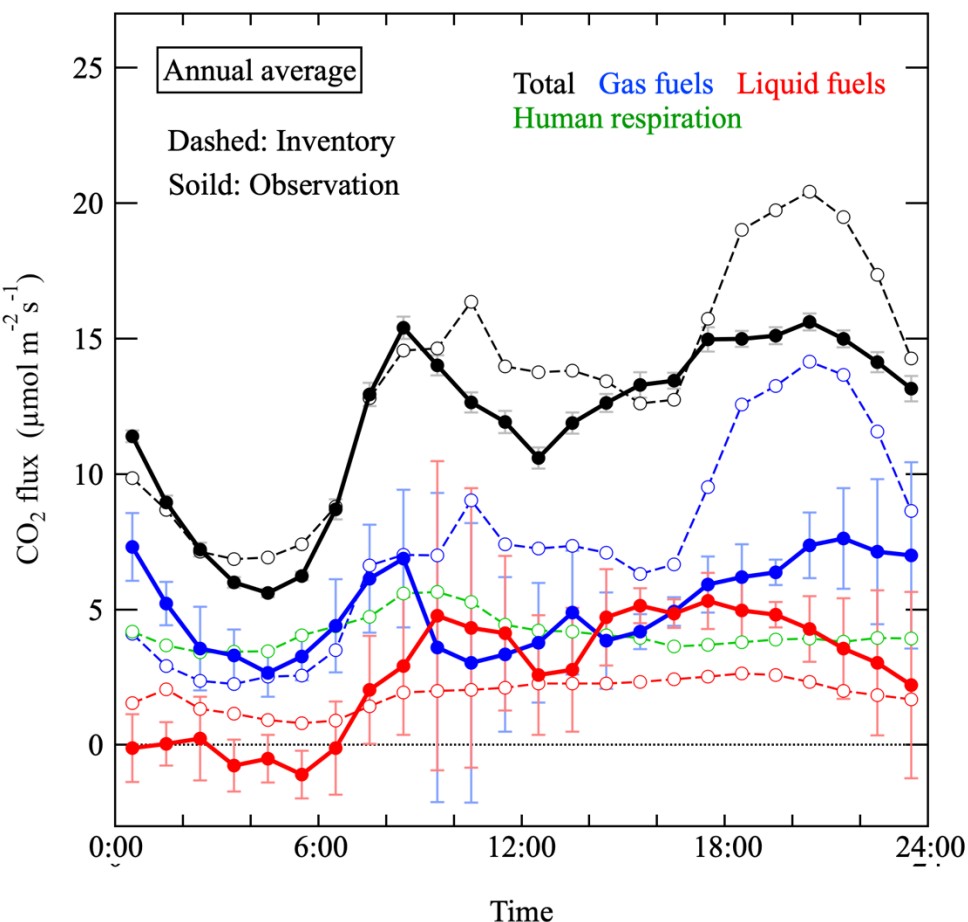

**Figure 10: Same as in Fig. 9 but for the annual average diurnal cycles. The error bars for the estimated CO$_2$ flux from liquid fuels consumption are the standard deviations of the diurnal cycles of the flux for respective seasons from the annual average cycle, assuming that the actual diurnal cycles of liquid fuels consumption do not change significantly throughout the year (see text).**
