# Peer review of "O2:CO2 exchange ratio for net turbulent flux observed in an Urban Area of Tokyo, Japan and its application to an evaluation of anthropogenic CO2 emissions"

_Atmospheric Chemistry and Physics, 2019_

## Referee Comment (RC1) · Anonymous Referee #2 · 27 Nov 2019

In this paper, the authors use continuous and simultaneous measurements of $O_2$ and $CO_2$ to determine the net oxidative ratio (OR) of atmospheric influences in a densely settled urban location. They then use an independent set of $CO_2$ flux measurements to calculate an eddy diffusion coefficient (K). With K in hand, they use vertical gradients in $O_2$ and $CO_2$ (in particular, differences in measured $O_2$ and $CO_2$ at two different heights on their sampling tower) to infer local fluxes of $O_2$ and $CO_2$, and from these, the net OR for fossil fuels. Assuming only two fossil fuel types (gaseous fuels and liquid gasoline), each with its own distinct OR, the combination of total emissions and net OR constrains

the relative fractions of the two fuel types.

Overall, the paper presents high-quality measurements that are definitely worthy of publication. The use of the high-low differences, in conjunction with the eddy-flux measurements of $CO_2$, is clever and yields valuable information about surface $O_2$ fluxes. Unfortunately, there is a significant problem with the interpretation of the measurements that needs to be addressed. In addition, there are many spots throughout the manuscript that require clarification.

**Primary scientific concern:**

One of the major challenges of working with tower data is determining the region of influence (the "footprint") for the tower. Calculations of the OR from $O_2 - CO_2$ covariation are particularly challenging, since the lower-frequency data from a paramagnetic analyzer lend themselves to aggregating data over extended periods. The authors acknowledge this in lines 133-135. However, the problem in this analysis is more profound than simply scaling footprints (inversely) by data-rate. This is because the OR slopes shown in Fig. 4 (lower panel) include data from the entire 18 month set of observations. Consequently, this is effectively a global average number with local influences superimposed.

To understand this, first consider a point in the plot with very low $O_2$ (and high $CO_2$). Maybe this parcel started with relatively high $O_2$ and was influenced by a great deal of local combustion. Or maybe it's part of an air mass that arrived from some distant location (highly influenced by combustion) and was relatively unaffected by local fluxes. Compare this to a point with relatively high $O_2$ (and low $CO_2$). If this point was measured hours before the low-$O_2$ one, and the wind pattern was roughly constant, chances are good that $O_2$ fell due to local combustion. In contrast, if this high-$O_2$ point was measured days (or months) before, it might have come from a totally different region and the difference from the first point reflects local influences to a much smaller degree. One solution is to choose much shorter aggregation periods when determining

$OR_{atm}$.

In short, all of the analysis of $OR_{atm}$, and the comparisons of $OR_{atm}$ with $OR_f$ need to be reconsidered.

For this reason, I will not comment further on the parts of the manuscript that involve the interpretation of $OR_{atm}$.

**Other scientific concerns:**

Line 30: In addition to Mitchell et al., please cite Sargent et al., PNAS 2018.

Lines 59-60: Does the vegetated area actually change seasonally? Or is it that the vegetation is mostly dormant in the winter?

Lines 71-72: As I understand it, the samples are measured with a paramagnetic analyzer relative to secondary standards. It's the secondary standards that are measured against the primary standard with a mass spectrometer. This is not what this sentence says.

Lines 73-75: Air is being drawn down at 10l/m and a very small subset of that airstream is being analyzed. There is no mention here of the possibility of fractionation at this sampling, of tests to detect fractionation, nor of measures to prevent it. This is something that Stephens et al. (DOI: 10.1175/JTECH1959.1 ) discusses extensively. Perhaps this is discussed in the original methods paper, but it should at least be mentioned here.

Lines 75-76: If air is measured first at one height, then the other, and air is measured for 10minutes at each height, isn't each measurement cycle 20minutes long (and thus, 9 cycles is 180 minutes)?

Line 79: How is a correction made for Ar? The paramagnetic analyzer doesn't measure this species. Again, this might be presented in 2014 Tellus paper, but a few words of explanation here would be welcome.

Line 83: Why are uncertainties being quoted for 30minute averages when atmospheric measurements are only made for 10-minute intervals on each intake, and standards are measured for 5-minute intervals.

Line 91: What does "span-difference" mean? Please clarify.

Lines 114-115: Downward excursions in $O_2$ may be due to consumption within the canopy, or non-local influences being transported to the tower. If they coincide with positive excursions in deltaO2, then I would be convinced that the cause is consumption within the local canopy, but until you show that the two excursions are coincident, you can't claim local consumption is the cause.

Line 154: If errors in both species are non-negligible, a standard least-squares linear regression will give the wrong slope. Instead a Deming regression is required (which reduces to an orthogonal fit in the case of equal areas).

Lines 183ff: A very basic back-of-the envelope calculation would be appropriate here to indicate whether human respiration really was utterly negligible or not. For example, the population density given for this area is 0.016 people $m^{-2}$. If each requires 2000 kcal/day, this could be supplied by metabolizing 3.34 moles of glucose, with a resulting consumption of $3.7 \mu mol m^{-2} s^{-1}$ of atmospheric $O_2$. This seems to be about 20% of the smallest values quoted on line 232: A modest, but non-negligible correction to the results presented here.

**Minor editorial comments:**

Line 44: Change to "In this paper, we first present the..."

Line 74: should read "and 37m was introduced"

Line 75: should read "100mL $min^{-1}$ with the pressure stabilized to 0.1 Pa and measured"

Line 85: should read "We used the gravimetrically prepared air-based"

Line 86: should read "1991) to determine"

Lines 87 and 90, "gravimetrically standard" should be replaced with "gravimetrically prepared standard"

Line 107: should read "activities. In contrast, the atmospheric$O_2$"

Line 111: should read "Therefore, we attribute the opposite phase" and "in this study mainly to fossil"

Line 124: Remove "by"

Line 131: End the sentence with "troposphere" and simply remove "whereas..."

Line 134: should read "1994). We note that"

Line 204: should read "standard error $(\sigma/\sqrt{n})$" (i.e. use symbols instead of writing it out).

Line 205: should read "negative values respectively, indicating"

Line 206: end the sentence with "the year." and remove "respectively".

Figure 6: There is no legend explaining the filled and unfilled symbols in the upper panel.

---

## Referee Comment (RC2) · Anonymous Referee #3 · 17 Dec 2019

In this paper, the authors use ∼18 months of continuous atmospheric CO2 and O2 measurements from an urban site near Tokyo in Japan. Using co-located CO2 eddy flux measurements to obtain a vertical diffusion coefficient and the CO2 and O2 concentration measurements from two heights, they obtain estimates of the local CO2 and O2 fluxes, and combine these to obtain the oxidative ratios of local fluxes (ORF) and the wider regional air masses (ORATM). They find ratios that are consistent with gaseous and liquid combustion fuel sources, with seasonal and diurnal variability consistent with wintertime heating and traffic signals respectively.

The use of CO2 eddy flux measurements with concentration measurements of CO2 and O2 to look at ORs in a forest environment has been already published by this group, however, this work is novel owing to the urban setting.

On the whole, the paper is well-written and well-presented, and the data appear to be of high quality, which is a worthy achievement for atmospheric O2. My main concern is the lack of met-related filtering of the atmospheric CO2 and O2 data prior to deriving the fluxes. I feel that the data handling as it is currently presented is perhaps too simplistic and should be taken further. I would like to see: a) filtering of the data to exclude periods that are highly influenced by regional not local fluxes (i.e. using associated met data, other tracers, or the concentration measurements themselves); b) more robust quantification of the ORs. While I can see the authors have attempted some robustness by calculating the ORs over two different time horizons (1-day and 1-week), I think this approach is not the best. Usually ORs are most robust during the onset of an atmospheric 'event' but not during the recovery phase when atmospheric conditions are unstable. So I would recommend only calculating ORs during the onset of atmospheric events. Also, a more robust approach to calculating ORs might consider other factors such as wind direction. This might also yield a more in-depth analysis of the OR results. I would also caution the authors about ascribing variations they see in the atmospheric ORs to changes in local fluxes, unless they can discount the influence of seasonal/diurnal atmospheric dynamic effects.

My specific comments are as follows:

Several times in the introduction, the authors mention that ORs can be used to separate out the contribution of different sources to the observed CO2 flux. I cannot think of a way this would work in reality without additional information (i.e. from bottom up inventories) unless one has a very idealised case with discrete sources coming from very different wind directions, for example. But for most cities, the sources are mixed. Ultimately, the measured OR will be a mixture of all the sources in the footprint, so it could be used to 'check' modelled OR estimates (although two 'wrongs' can also make

a 'right'), but it cannot be used in itself to distinguish $CO_2$ fluxes from different sources.

Several times, the flux footprint vegetated area is stated at 9% in summer and 2% in winter. The authors should state how these values are derived. They also seem too low, based on the images given in figure 1. If there is a strong seasonal difference in the footprint of the measurements between summer and winter, how are the authors sure that the OR results they obtain are related to changing flux patterns/behaviour and not simply caused by the changing footprint?

Lines 73-74: How did the authors subsample from such a high flowrate without using a tee and causing fractionation of $O_2$ wrt $N_2$? Please clarify, since this is an important technical point.

Lines 91-96: I'm not sure that the logic is valid here. Since ORs are calculated from regressing two sets of data, they are most sensitive to inaccuracies at the high/low ends of the scale. So I think the authors might find the uncertainties in OR are larger than the 1% uncertainties at the high end of the $CO_2$ scale. The easiest way to check is to recalculate some ORs using a 1% difference in the high $CO_2$ values and see how large the difference in OR is. My guess would be it's more like 10%.

Lines 115-116: two things here. Firstly, I would caution against attributing changes in the atmospheric data to changes in fuel usage without very strong evidence, ideally from multiple sources. Such changes can sometimes be caused by seasonal changes in atmospheric dynamics or changing footprint, see my comment above. Secondly, winter is usually associated with more boundary layer turbulence, not more stratification. If the authors disagree, please can they provide a citation to back up this statement, which seems to me to be erroneous.

Lines 177-178: I think the authors state here that there was no coal fluxes observed because no ORs were 1.17? If so, I would strongly advise the authors retract this statement, since it is very possible that a mixture of coal and gas could give a ratio that looks like liquid fuel, and yet perhaps there was no liquid fuel burnt at the time. If there

Interactive
comment

is independent evidence for expecting no or very little coal (such as from an inventory) then please provide this here.

Line 194: "on the other hand" used twice in same paragraph. Suggest to rewrite one of them. Or better still to omit entirely, since this is rather colloquial language for such a publication.

Line 202: Suggest to rewrite "seasonal "climatological" diurnal cycles" as I am not sure what the authors mean. I think what is meant is the average diurnal cycle in different seasons.

Lines 219, 221: I would advise caution again here, unless there is independent evidence to back these statements up. It would also be nice to see how much diurnal variation there is in the site footprint, in addition to the seasonal variation.

Lines 237-247 and corresponding text in conclusions: I do not see the value in this paragraph or it's relevance to the rest of the paper. The authors state that the O2 urban fluxes are very large compared to the global mean O2 fluxes, but the global O2 decrease accounts for O2 fluxes from all urban regions, so I'm not sure what the point of the comparison is. And as the authors themselves state, it is unrealistic that urban O2 depletion would lead to atmospheric O2 falling to levels that are dangerous for human health (perhaps this is possible for isolated indoor environments, but not in the free atmosphere – this has been debunked many times now by many people). I would recommend the authors remove this paragraph and focus solely on the OR analyses.

Figure 2: It is hard to see the seasonal difference of delta O2 and delta CO2 with the current y-axis scaling.

Figure 3: please separate the O2 and CO2 grey data points with more white space so the two time series datasets can be viewed independently/more easily.

Figure 4: do these regression fits account for the difference in measurement precision between CO2 and O2? Also, please state whether the fits account for both x and y

uncertainties.

Figure 6: what are the open circles? The evening peak seems to occur too late in the day to be accounted for by traffic alone (especially in winter). Also, this peak is much broader than the morning peak, suggesting there is a net flux of traffic out of the region over time (whereas presumably this is not the case). I think some more in-depth analysis into these patterns would be useful here.

[Figure]

---

## Editor Comment (EC1) · Thomas Karl (Editor) · 11 Mar 2020

Concerning the treatment of urban fluxes, issues about proper QAQC treatment of eddy covariance fluxes have been raised. It is noted that (depending on the application) eddy covariance fluxes should generally be filtered according to several well established criteria (e.g. stationarity, u*, stabilitiy). An appropriate method can for example be found in Lee et al. Handbook of Micrometeorology (ISBN 978-1-4020-2265-4). Chapter 9. Foken et al., Post field data quality control.

[Figure]

Another important issue for urban flux calculations is the treatment of sonic anemometer rotation (e.g. planar fit, or double rotation depedning on the application). These rotations can be quite sector dependent in urban areas (see for example https://www.atmos-meas-tech-discuss.net/amt-2019-272/amt-2019-272.pdf). It should be noted which rotation was used, and whether it was applied sector dependent.

Flux footprint: Neftel et al. based their footprint model on Horst and Weil, and evaluated the footprint model specificially for grassland, which has a completely different surface characteristic than an urban landcover. While there is currently no true parameterization for the urban roughness layer, an updated footprint model by Kljun et al. (https://www.geosci-model-dev.net/8/3695/2015/), that was developed for tall canopies, would perhaps give a better representation of the urban flux footprint.

---

## Author Comment (AC3) · 23 Apr 2020

**Responses to the Editor:**

Thank you very much for your significant and useful comments on the paper "$O_2$:$CO_2$ exchange ratio for net turbulent flux observed in an Urban Area of Tokyo, Japan and its application to an evaluation of anthropogenic $CO_2$ emissions" by Ishidoya et al. We have revised the manuscript, considering your comments and suggestions. Details of our revision are as follows;

**1) Concerning the treatment of urban fluxes, issues about proper QAQC treatment of eddy covariance fluxes have been raised. It is noted that (depending on the application) eddy covariance fluxes should generally be filtered according to several well established criteria (e.g. stationarity, u\*, stabilitiy). An appropriate method can for example be found in Lee et al. Handbook of Micrometeorology (ISBN 978-1-4020-2265-4). Chapter 9. Foken et al., Post field data quality control.**

Lines 115-116: The sentences have been added to describe the QAQC treatment of eddy covariance fluxes. Measurement runs with good quality were selected based on the flag calculated in Eddypro software. We used the runs with flag 0 - 2 those are at least suitable for general analysis such as annual budgets.

**2) Another important issue for urban flux calculations is the treatment of sonic anemometer rotation (e.g. planar fit, or double rotation depending on the application). These rotations can be quite sector dependent in urban areas (see for example https://www.atmos-meas-tech-discuss.net/amt-2019-272/amt-2019-272.pdf). It should be noted which rotation was used, and whether it was applied sector dependent.**

Line 114: We used "double rotation" in the flux calculation. The words "by using the double rotation algorithm" have been added to the sentence.

**3) Flux footprint: Neftel et al. based their footprint model on Horst and Weil, and evaluated the footprint model specifically for grassland, which has a completely different surface characteristic than an urban landcover. While there is currently no true parameterization for the urban roughness layer, an updated footprint model by Kljun et al. (https://www.geosci-model-dev.net/8/3695/2015/), that was developed for tall canopies, would perhaps give a better representation of the urban flux footprint.**

As you pointed out, the model of Kljun et al. (2015) could be more appropriate. However, we consider the model of Neftel et al. (2008) is also applicable in the present study due to two technical reasons; 1) the turbulent sensors were installed at about 5 times higher than the mean building height, as added in the text (Lines 111-112). Therefore, our turbulent measurements were taken outside the roughness sublayer which is about 2 times higher than the canopy height (Cheng and Castro, 2002). The model of Neftel et al. could be valid in this situation. 2) We used the footprint model to capture roughly the source area of flux. If we have attempted detailed analysis like Christen et al. (2011), which made emission modeling with the footprint model, the model of Kljun et al. should be better. We calculated the ratio of greens in the footprint, however the ratio was very small value and the variation due to model type would also be small.

References

Cheng H, Castro IP (2002) Near wall flow over urban-like roughness. Bound Layer Meteorol 104:229–259.

Christen, A., et al. (2011). Validation of modeled carbon-dioxide emissions from an urban neighborhood with direct eddy-covariance measurements. Atmospheric Environment, 45(33), 6057– 6069. https://doi.org/10.1016/j.atmosenv.2011.07.040.

---

## Author Response (AR1)

**Responses to Referee 2:**

Thank you very much for your significant and useful comments on the paper " $O_2$ :CO2 exchange ratio for net turbulent flux observed in an Urban Area of Tokyo, Japan and its application to an evaluation of anthropogenic CO2 emissions" by Ishidoya et al. The title of the paper has been changed from the ACPD paper. We have revised the manuscript, considering your comments and suggestions. Details of our revision are as follows;

**Primary scientific concern**

One of the major challenges of working with tower data is determining the region of "footprint") influence (the for the tower. Calculations of the OR  $fromO_2$ -CO2covariation are particularly challenging, since the lower-frequency data from a paramagnetic analyzer lend themselves to aggregating data over extended periods. The authors acknowledge this in lines 133-135. However, the problem in this analysis is more profound than simply scaling footprints (inversely) by data-rate. This is because the OR slopes shown in Fig. 4 (lower panel) include data from the entire 18 month set of observations. Consequently, this is effectively a global average number with local influences superimposed.

To understand this, first consider a point in the plot with very low O2 (and high CO2). Maybe this parcel started with relatively high O2 and was influenced by a great deal of local combustion. OR maybe it's part of an air mass that arrived from some dis- tant location (highly influenced by combustion) and was relatively unaffected by local fluxes. Compare this to a point with relatively high O2 (and low CO2). If this point was measured hours before the low-O2 one, and the wind pattern was roughly constant, chances are good that O2 fell due to local combustion. In contrast, if this high-O2 point was measured days (or months) before, it might have come from a totally different re gion and the difference from the first point reflects local influences to a much smaller degree. One solution is to choose much shorter aggregation periods when determining ORatm.

In short, all of the analysis of ORatm, and the comparisons of ORatm with ORF need to be reconsidered.

For this reason, I will not comment further on the parts of the manuscript that involve

**the interpretation of ORatm.**

Considering your comments, we have reconsidered the comparison of  $OR_{atm}$  with  $OR_{F}$ . We have added the sentence not only to state the problem to use  $OR_{atm}$  but also to clarify the purpose of the comparison (line 185-190). The comparisons by choosing 12-hour aggregation period have also been added (line 191-210 and Fig. 6). The discussion for the comparisons by choosing 1-week aggregation period has been modified, and we have concluded to use  $OR_F$  rather than  $OR_{atm}$  is more appropriate to validate inventory-based  $CO_2$  emissions from gas, liquid and solid fuels in the flux footprint (line 211-252). Moreover, we have newly added discussion to estimate the average diurnal cycles of  $CO_2$  fluxes from gas and liquid fuels consumption separately by using the  $OR_F$ ,  $CO_2$  flux, and inventory-based  $CO_2$  emissions from gas consumption and traffic (line 290-344, Fig. 9 and Fig. 10). The inventory-based emission data have been updated from Hirano et al. (2015) for the present study.

**Other scientific concerns:**

**1) Line 30: In addition to Mitchell et al., please cite Sargent et al., PNAS 2018.**

Line 34: We have cited Sargent et al. (2018), as suggested.

**2) Lines 59-60: Does the vegetated area actually change seasonally? Or is it that the vegetation is mostly dormant in the winter?**

Line 66: The sentence has been modified as "The flux footprint includes vegetated area of 9% in the summer and 2% in the winter, reflecting seasonal changes in the wind direction." As seen in Fig. 1, the vegetated area included in the flux footprint actually change seasonally due to seasonal changes in the wind direction. It is noted that calculation of the flux footprint has been updated by using the model of Neftel et al. (2008) (line 63).

3) Lines 71-72: As I understand it, the samples are measured with a paramagnetic analyzer relative to secondary standards. It's the secondary standards that are measured against the primary standard with a mass spectrometer. This is not what this sentence says.

Lines 76-79: That is as you pointed out. We have changed the phrase as "In this study,  $\delta(O_2/N_2)$  values of each air sample were measured with the paramagnetic analyzer using working standard air that was measured against our primary standard air (Cylinder No. CRC00045; AIST-scale) using a mass spectrometer (Thermo Scientific Delta-V) (Ishidoya and Murayama, 2014)."

4) Lines 73-75: Air is being drawn down at 101/m and a very small subset of that airstream is being analyzed. There is no mention here of the possibility of fractionation at this sampling, of tests to detect fractionation, nor of measures to prevent it. This is something that Stephens et al. (DOI: 10.1175/JTECH1959.1) discusses extensively. Perhaps this is discussed in the original methods paper, but it should at least be mentioned here.

Lines 80-87: We have added the sentences to discuss the possible fractionation for the measurements in this study.

5) Lines 75-76: If air is measured first at one height, then the other, and air is measured for 10minutes at each height, isn't each measurement cycle 20minutes long (and thus, 9 cycles is 180 minutes)?

Lines 87-88: The phrase "After 9 measurement cycles (90 minutes)" has been changed to "After 9 cycles of measurements (5 and 4 cycles for 37 and 52 m, respectively)" to clarify the meaning.

6) Line 79: How is a correction made for Ar? The paramagnetic analyzer doesn't measure this species. Again, this might be presented in 2014 Tellus paper, but a few words of explanation here would be welcome.

Lines 90-93: We have modified the sentence as follows to explain the correction method briefly "The dilution effects on the  $O_2$  mole fraction measured by the paramagnetic analyzer were corrected experimentally, not only for the changes in  $CO_2$  of the sample air or standard gas measured by the NDIR, but also for the changes in Ar of the standard gas measured by the mass spectrometer as  $\delta(Ar/N_2)$ ."

**7) Line 83: Why are uncertainties being quoted for 30minute averages when atmospheric measurements are only made for 10-minute intervals on each intake, and standards are measured for 5-minute intervals.**

Lines 93-94: The sentence has been modified to show the analytical reproducibility for 2-minute average only.

**8) Line 91: What does "span-difference" mean? Please clarify.**

Lines 102-105: We avoid to use the word "span-difference" and changed the sentence as follows "Although the highest  $CO_2$  concentration of the gravimetrically standard of the NIES-09 scale is similar to that of the TU-10 scale, a slope of 0.974 ppm ppm-1 is derived from a least-squares regression line fitted to the relationship between the  $CO_2$ concentrations observed by NDIR on the TU-10 scale and those by CRDS on the NIES-09 scale with a correlation coefficient (r) of 0.978.".

9) Lines 114-115: Downward excursions in  $O_2$  may be due to consumption within the canopy, or non-local influences being transported to the tower. If they coincide with positive excursions in delta $O_2$ , then I would be convinced that the cause is consumption within the local canopy, but until you show that the two excursions are coincident, you can't claim local consumption is the cause.

Lines 127-132 and Fig. 3: The sentences and figures have been added to show the two excursions are coincident.

10) Line 154: If errors in both species are non-negligible, a standard least-squares linear regression will give the wrong slope. Instead a Deming regression is required (which reduces to an orthogonal fit in the case of equal areas).

Lines 170-178: We have changed the regression method to Deming regression throughout the paper for calculating OR, as suggested.

11) Lines 183ff: A very basic back-of-the envelope calculation would be appropriate here to indicate whether human respiration really was utterly negligible or not. For example, the population density given for this area is 0.016 people m-2. If each requires 2000 kcal/day, this could be supplied by metabolizing 3.34 moles of glucose, with a resulting consumption of  $3.7\mu$ molm-2s-1 of atmospheric O2. This seems to be about

**20% of the smallest values quoted on line 232: A modest, but non-negligible correction to the results presented here.**

Lines 290-344, Fig. 9 and Fig. 10: We have added discussion to estimate the average diurnal cycles of  $CO_2$  fluxes from gas and liquid fuels consumption separately by using the  $OR_F$ ,  $CO_2$  flux, and inventory-based  $CO_2$  emission from human respiration, in order to validate the inventory-based  $CO_2$  emissions from gas consumption and traffic. The inventory-based  $CO_2$  emission from human respiration is close to the value in your comments.

**Minor editorial comments:**

**1) Line 44: Change to "In this paper, we first present the. . ."**

Line 48: The words "In this paper, we present firstly the..." have been changed to "In this paper, we first present the...", as suggested.

**2) Line 74: should read "and 37m was introduced"**

Lines 80-95: The sentences, including the words that were pointed out, have been rewritten.

**3) Line 75: should read "100mL min-1 with the pressure stabilized to 0.1 Pa and measured"**

Lines 83-84: The words have been changed, as suggested.

**4) Line 85: should read "We used the gravimetrically prepared air-based"**

Line 96: The words have been changed, as suggested.

**5) Line 86: should read "1991) to determine"**

Line 97: The words have been changed, as suggested.

**6) Lines 87 and 90, "gravimetrically standard" should be replaced with "gravimetrically prepared standard"**

Lines 98 and 101: The words "gravimetrically standard" have been replaced with "gravimetrically prepared standard".

**7) Line 107: should read "activities. In contrast, the atmosphericO2"**

Line 119: The words "On the other hand" have been changed to "In contrast", as suggested.

**8) Line 111: should read "Therefore, we attribute the opposite phase" and "in this study mainly to fossil"**

Lines 123-124: The sentence has been modified as suggested.

**9) Line 124: Remove "by"**

Line 140: The word "by" has been removed.

**10)** Line 131: End the sentence with "troposphere" and simply remove "whereas..." Line 147: The words "in the troposphere, whereas it is..." have been changed to "in the troposphere. It is...".**

**11) Line 134: should read "1994). We note that"**

Line 150: The words "1994). It is noted" have been changed to "1994). We note that".

**12) Line 204: should read "standard error $(\sigma/\sqrt{n})$ " (i.e. use symbols instead of writing it out).**

Line 258: The words have been changed, as suggested.

**13) Line 205: should read "negative values respectively, indicating"**

Lines 258-259: The words have been changed, as suggested.

**14) Line 206: end the sentence with "the year." and remove "respectively".**

Line 260: The words have been changed, as suggested.

**15) Figure 6: There is no legend explaining the filled and unfilled symbols in the upper panel.**

Figure 8: The words to explain the filled and unfilled circles in the upper panel have been added to the figure caption. It is noted the number of the figure has been changed from that in the ACPD paper.

**Responses to Referee 3:**

Thank you very much for your significant and useful comments on the paper " $O_2$ :CO2 exchange ratio for net turbulent flux observed in an Urban Area of Tokyo, Japan and its application to an evaluation of anthropogenic CO2 emissions" by Ishidoya et al. The title of the paper has been changed from the ACPD paper. We have revised the manuscript, considering your comments and suggestions. Details of our revision are as follows;

**Main concern:**

My main concern is the lack of met-related filtering of the atmospheric CO2 and O2 data prior to deriving the fluxes. I feel that the data handling as it is currently presented is perhaps too simplistic and should be taken further. I would like to see: a) filtering of the data to exclude periods that are highly influenced by regional not local fluxes (i.e. using associated met data, other tracers, or the concentration measurements themselves); b) more robust quantification of the ORs. While I can see the authors have attempted some robustness by calculating the ORs over two different time horizons (1-day and 1-week), I think this approach is not the best. Usually ORs are most robust during the onset of an atmospheric 'event' but not during the recovery phase when atmospheric conditions are unstable. So I would recommend only calculating ORs during the onset of atmospheric events. Also, a more robust approach to calculating ORs might consider other factors such as wind direction. This might also yield a more in-depth analysis of the OR results. I would also caution the authors about ascribing variations they see in the atmospheric ORs to changes in local fluxes, unless they can discount the influence of seasonal/diurnal atmospheric dynamic effects.

Considering your comments, we have added the discussion including filtering of the data using wind direction (line 191-210 and Fig. 6). For the analyses of specific events, we have reported the OR values and simultaneously-measured  $PM_{2.5}$  aerosol composition for a week-long pollution event by Kaneyasu et al. (2020) (line 208-210). Considering the results of the discussion, we decide to use all the O2 and CO2 concentration data without filtering by the wind direction, to increase the number of data

points for calculating  $OR_F$  and  $OR_{atm}$ ; this is consistent with the purpose of this study to derive representative OR values at the YYG site in order to validate the  $CO_2$  emission inventory updated from Hirano et al. (2015). It is noted that we have newly added discussion to estimate the average diurnal cycles of  $CO_2$  fluxes from gas and liquid fuels consumption separately by using the  $OR_F$ ,  $CO_2$  flux, and inventory-based  $CO_2$  emission from human respiration, in order to validate the inventory-based  $CO_2$  emissions from gas consumption and traffic (line 290-344, Fig. 9 and Fig. 10).

**Specific comments:**

1) Several times in the introduction, the authors mention that ORs can be used to separate out the contribution of different sources to the observed CO2 flux. I cannot think of a way this would work in reality without additional information (i.e. from bottom up inventories) unless one has a very idealised case with discrete sources coming from very different wind directions, for example. But for most cities, the sources are mixed. Ultimately, the measured OR will be a mixture of all the sources in the footprint, so it could be used to 'check' modelled OR estimates (although two 'wrongs' can also make a 'right'), but it cannot be used in itself to distinguish CO2 fluxes from different sources.

We agree with you that OR can be used to check modelled OR but cannot be used in itself to distinguish CO2 fluxes from different sources. Therefore, we have changed some sentences, e.g. from "…then the information can be used to separate out the contributions of the gaseous, liquid, and solid fuels, and the terrestrial biospheric activities to the observed CO2 flux" to "…then such information can be used as a useful constraint for evaluating the contributions of the gaseous, liquid, and solid fuels, and the terrestrial biospheric activities to the observed CO2 flux" to "…then such information can be used as a useful constraint for evaluating the contributions of the gaseous, liquid, and solid fuels, and the terrestrial biospheric activities to the observed CO2 flux" (line 44-46). Moreover, as mentioned above, we have newly added discussion to estimate the average diurnal cycles of CO2 fluxes from gas and liquid fuels consumption separately by using the ORF, CO2 flux, and inventory-based CO2 emission from human respiration, in order to validate the inventory-based CO2 emissions from gas consumption and traffic (line 290-344, Fig. 9 and Fig. 10).

Several times, the flux footprint vegetated area is stated at 9% in summer and 2% in winter. The authors should state how these values are derived. They also seem too low, based on the images given in figure 1. If there is a strong seasonal difference in the footprint of the measurements between summer and winter, how are the authors sure that the OR results they obtain are related to changing flux patterns/behaviour and not simply caused by the changing footprint?

The vegetation area was calculated from the area included in the aerial photo in Fig. 1 by considering the contribution to flux. The calculation method for the footprint is based on the model of Neftel et al. (2008) (line 63). It is noted the footprint and the caption of Fig. 1 have been revised. As you pointed out, there is a seasonal difference in the footprint between summer and winter due to the seasonal difference of the prevailing direction of wind. However, as shown in the contour lines in Fig. 1, which indicate contribution in measured flux (60, 50, 40, 30, 20 and 10% from outside to inside), the dominant contribution to flux is from the adjacent area of the observation tower; within about 300 m from the tower in both seasons. Land cover is nearly uniform in this dominant footprint area as shown in Fig.1. Therefore, the observed  $OR_F$  values are determined mainly by the  $O_2$  and  $CO_2$  fluxes from the urban area and the effect of seasonal difference in the footprint to the  $OR_F$  would be relatively small.

**2) Lines 73-74: How did the authors subsample from such a high flowrate without using a tee and causing fractionation of $O_2$ wrt $N_2$ ? Please clarify, since this is an important technical point.**

Lines 80-87: We have added the sentences to show the subsampling method and discuss the possible fractionation for the measurements in this study.

3) Lines 91-96: I'm not sure that the logic is valid here. Since ORs are calculated from regressing two sets of data, they are most sensitive to inaccuracies at the high/low ends of the scale. So I think the authors might find the uncertainties in OR are larger than the 1% uncertainties at the high end of the CO2 scale. The easiest way to check is to recalculate some ORs using a 1% difference in the high CO2 values and see how large the difference in OR is. My guess would be it's more like 10%.

Lines 101-108: The OR is calculated as a ratio of difference of  $O_2$  concentration to that of  $CO_2$  concentration, so that we consider the effect of the span-difference of  $CO_2$  on

| O 2 anomaly
High (ppm) | O 2 anomaly
Low (ppm) | CO 2 on scale-1
High (ppm) | CO 2 on scale-1
Low (ppm) | OR |
|--------------------------------------|-------------------------------------|------------------------------------------|-----------------------------------------|----|
| -400                                 | -600                                | 600                                      | 500                                     | 2  |
| -400                                 | -600                                | 400                                      | 300                                     | 2  |

the OR does not depend on the absolute value of the CO2 concentration, as following idealized tables:

| O 2 anomaly | O 2 anomaly | CO 2 on scale-2 | CO 2 on scale-2 | OR   |
|------------------------|------------------------|----------------------------|----------------------------|------|
| High (ppm)             | Low (ppm)              | High (ppm)                 | Low (ppm)                  |      |
| -400                   | -600                   | 612                        | 510                        | 1.96 |
| -400                   | -600                   | 408                        | 306                        | 1.96 |

\*OR values are calculated by " $-(O_2_high - O_2_low)/(CO_2_high - CO_2_low)$ ", and the span-difference of CO2 between scale-1 and scale-2 is 2%.

We have also modified the sentences and allowed the uncertainty of within 3% for OR, which is larger than the ACPD, due to the span-uncertainties of  $O_2$  and  $CO_2$  concentrations.

4) Lines 115-116: two things here. Firstly, I would caution against attributing changes in the atmospheric data to changes in fuel usage without very strong evidence, ideally from multiple sources. Such changes can sometimes be caused by seasonal changes in atmospheric dynamics or changing footprint, see my comment above. Secondly, winter is usually associated with more boundary layer turbulence, not more stratification. If the authors disagree, please can they provide a citation to back up this statement, which seems to me to be erroneous.

As already mentioned above, we have newly added discussion using the  $CO_2$  emission inventory data of gas consumption, traffic and human respiration around YYG to show the evidence for changes in fuel usage (line 290-344, Fig. 9 and Fig. 10). The sentences and figures have been added to show the evidence that  $O_2$  is consumed within the urban canopy at YYG especially in winter (line 127-132 and Fig. 3), and the words "a more stable stratification of surface atmosphere" have been changed to "a temperature inversion near the surface" to make the meaning clearer. It should be note here that the stable layer can be found mainly in winter (Kanda et al., 2005), meaning less turbulence in winter than in summer.

M. Kanda, R. Moriwaki and Y. Kimoto, Temperature Profiles Within and Above an Urban Canopy, Boundary-Layer Meteorology volume 115, 499–506, 2005.

5) Lines 177-178: I think the authors state here that there was no coal fluxes observed because no ORs were 1.17? If so, I would strongly advise the authors retract this statement, since it is very possible that a mixture of coal and gas could give a ratio that looks like liquid fuel, and yet perhaps there was no liquid fuel burnt at the time. If there is independent evidence for expecting no or very little coal (such as from an inventory) then please provide this here.

Lines 221-225: The sentences have been added to show independent evidence for expecting very small contributions of coal.

6) Line 194: "on the other hand" used twice in same paragraph. Suggest to rewrite one of them. Or better still to omit entirely, since this is rather colloquial language for such a publication.

Lines 239-242: The words "on the other hand" were removed, as suggested.

7) Line 202: Suggest to rewrite "seasonal "climatological" diurnal cycles" as I am not sure what the authors mean. I think what is meant is the average diurnal cycle in different seasons.

Line 254: The words "seasonal "climatological" diurnal cycles" have been changed to "average diurnal cycles".

8) Lines 219, 221: I would advise caution again here, unless there is independent evidence to back these statements up. It would also be nice to see how much diurnal variation there is in the site footprint, in addition to the seasonal variation.

As noted in our response to your comment No.1, the flux footprint was mainly located around the tower. The footprint had diurnal variation in its location, however it was still located in the relatively homogeneous area around the tower.

Considering your comments, we made some revision in our manuscript (lines 271-273, 290-344, Fig. 9 and Fig. 10). We have modified the sentences considering your comments, and we have added sentences and figures to discuss the estimations of the average diurnal cycles of  $CO_2$  fluxes from gas and liquid fuels consumption separately by using the  $OR_F$ ,  $CO_2$  flux, and inventory-based  $CO_2$  emission from human respiration, in order to validate the inventory-based  $CO_2$  emissions from gas consumption and traffic. The inventory-based emission data have been updated from Hirano et al. (2015) for the present study. We hope these revisions will meet your suggestion to show independent evidence to back the statements up.

9) Lines 237-247 and corresponding text in conclusions: I do not see the value in this paragraph or it's relevance to the rest of the paper. The authors state that the  $O_2$  urban fluxes are very large compared to the global mean  $O_2$  fluxes, but the global  $O_2$  decrease accounts for  $O_2$  fluxes from all urban regions, so I'm not sure what the point of the comparison is. And as the authors themselves state, it is unrealistic that urban  $O_2$  depletion would lead to atmospheric  $O_2$  falling to levels that are dangerous for human health (perhaps this is possible for isolated indoor environments, but not in the free atmosphere – this has been debunked many times now by many people). I would recommend the authors remove this paragraph and focus solely on the OR analyses.

Lines 284-289: We agree with you that the statements in the paragraph do not have enough value to discuss in detail. Therefore, the sentences have been much shortened and started the phrase "In this regard..." to clarify that is just for reference, and we have focused on the OR analyses combined with the inventory-based  $CO_2$  emissions prepared for the present study (line 290-344, Fig. 9 and Fig. 10).

**10) Figure 2: It is hard to see the seasonal difference of delta O2 and delta CO2 with the current y-axis scaling.**

Figure 3: We have added the figure to show the  $O_2$  and  $CO_2$  concentrations,  $\Delta O_2$  and  $\Delta CO_2$  for the period December 16 – 23 and July 1 – 9, 2016, to see the seasonal difference.

11) Figure 3: please separate the O2 and CO2 grey data points with more white space so the two time series datasets can be viewed independently/more easily.

Figure 4: The figure has been modified, as suggested. It is noted the number of the figure has been changed from that in the ACPD paper.

12) Figure 4: do these regression fits account for the difference in measurement precision between  $CO_2$  and  $O_2$ ? Also, please state whether the fits account for both x and y uncertainties.

Lines 170-178: We have changed the regression method to Deming regression throughout the paper for calculating OR, in order to account not only for the difference in measurement precision between  $CO_2$  and  $O_2$  but also for both x and y uncertainties.

13) Figure 6: what are the open circles? The evening peak seems to occur too late in the day to be accounted for by traffic alone (especially in winter). Also, this peak is much broader than the morning peak, suggesting there is a net flux of traffic out of the region over time (whereas presumably this is not the case). I think some more in-depth analysis into these patterns would be useful here.

Figure 8: The words to explain the filled and open circles in the upper panel have been added to the figure caption. It is noted the number of the figure has been changed from that in the ACPD paper. As to the detail analyses of the morning and evening peaks, we have added the OR analyses combined with the inventory-based  $CO_2$  emissions as mentioned above (line 290-344, Fig. 9 and Fig. 10).

**O2:CO2 exchange ratio for net turbulent flux observed in an Urban Area of Tokyo, Japan and its application to an evaluation of anthropogenic CO2 emissions**

Shigeyuki Ishidova1, Hirofumi Sugawara2, Yukio Terao3, Naoki Kaneyasu1, 5 Nobuyuki Aoki1, Kazuhiro Tsuboi4, and Hiroaki Kondo1

1National Institute of Advanced Industrial Science and Technology (AIST), Tsukuba 305-8569, Japan 2Department of Earth and Ocean Sciences, National Defense Academy of Japan, Yokosuka 239-8686, Japan 3National Institute for Environmental Studies, Tsukuba 305-8506, Japan

4Meteorological Research Institute, Tsukuba 305-0052, Japan 10

Correspondence to: Shigevuki Ishidova (s-ishidova@aist.go.jp)

Abstract. In order to examine  $O_2$  consumption and  $CO_2$  emission in a megacity, continuous observations of atmospheric  $O_2$ and CO2 concentrations, along with CO2 flux, have been carried out simultaneously since March 2016 at the Yoyogi (YYG) site located in the middle of Tokyo, Japan. An average  $O_2$ :CO2 exchange ratio for net turbulent  $O_2$  and CO2 fluxes (ORF)

- between the urban area and the overlying atmosphere was obtained based on an aerodynamic method using the observed O2 15 and  $CO_2$  concentrations. The yearly mean  $OR_F$  was found to be 1.62, falling within the range of the average OR values of liquid and gas fuels, and the annual average daily mean  $O_2$  flux at YYG was estimated to be -16.3 µmol m-2s-1 based on the ORF and CO2 flux. By using the observed ORF and CO2 flux, along with the inventory-based CO2 emission from human respiration, we estimated the average diurnal cycles of CO2 fluxes from gas and liquid fuels consumption separately for each
- 20 season. Both the estimated and the inventory-based CO2 fluxes from gas fuels consumption showed average diurnal cycles with two peaks, one in the morning and another one in the evening; however, the evening peak of the inventory-based gas consumption was much larger than that estimated from the CO2 flux. This can explain the discrepancy between the observed and the inventory-based total CO2 flux at YYG. Therefore, simultaneous observations of ORF and CO2 flux are useful in validating CO2 
[revised manuscript text omitted]
 ORF and ORatm values in the summer were lower than ORff in Tokyo (1.65), but ORatm was also found to be lower than ORff for the Kanto area (1.52). These lower ORF and ORatm values, compared to those of the ORff suggest that the ratio of fossil fuel combustion to terrestrial biospheric activities and human
- respiration is lower in the summer than that in the winter. The slightly lower  $OR_{atm}$  than  $OR_F$  at YYG throughout the year is probably due to the higher contribution of the air mass from Kanto area to  $OR_{atm}$  than  $OR_F$ , since the Kanto area as a whole has lower  $OR_{ff}$  than for Tokyo; in addition, the south Kanto area (including Tokyo) has a larger vegetation coverage of about 50% than that in the area around YYG site. From the comparison results of the  $OR_F$  with  $OR_{atm}$  in Fig. 5 – 7, it is suggested that the  $OR_{atm}$  reflects wider footprints of  $O_2$  and  $CO_2$  than  $OR_F$  for the aggregation periods at least longer than 12 hours to
- 250 calculate the ORatm. Therefore, to use ORF rather than ORatm is more appropriate to validate inventory-based CO2 emissions from gas, liquid and solid fuels in the flux footprint.

**3.3 Consumption of gas and liquid fuels estimated from the observed CO2 flux and O2:CO2 exchange ratio for net turbulent flux**

- In this section, we derive average diurnal cycles of ORF, CO2 and O2 flux and estimate the CO2 fluxes from gas and liquid 255 fuels consumption separately. Figure 8 shows the average diurnal cycles of  $\Delta$ O2 and  $\Delta$ CO2 for each season. To derive the average diurnal cycles, the observed  $\Delta$ O2 and  $\Delta$ CO2 values of each day in a season were overlaid on top of the values of other days, added up and divided by the number of days in the season. The error bars shown in Fig. 8 indicate ±1 standard error ( $\sigma/\sqrt{n}$ ). The  $\Delta$ O2 and  $\Delta$ CO2 values vary systematically in opposite phase and take positive and negative values respectively, indicating transport of O2 uptake and CO2 emission signals from the urban area to the overlying atmosphere
- 260 throughout the year. Daily maxima of  $\Delta O_2$  shown in Fig. 8 are higher in the winter than in the summer and occur in the nighttime. These characteristics would be attributable to an enhancement of the anthropogenic  $O_2$  consumption in the winter, while the nighttime decrease of  $O_2$  concentration would be due to the  $O_2$  consumption near the surface and temperature inversion near the surface. It must be noted that the  $\Delta CO_2$  values in the daytime are nearly zero, while the  $\Delta O_2$  values are not. The intercepts of the regression lines fitted to the relationship between  $\Delta O_2$  and  $\Delta CO_2$  in Fig. 8 are 0.27, 0.41, 0.45 and 0.44
- 265  $\mu$ mol mol-1 in DJF, MAM, JJA and SON, respectively. Unfortunately, we did not fix the cause(s) of such biases yet, although it may be related, to some extent, to natural exchange processes between the urban area and the overlying atmosphere. Therefore, because of these issues, the use of ORF, calculated by applying a Deming regression fitted to 2-hour period values of  $\Delta$ O2 and  $\Delta$ CO2 of the climatological diurnal cycle (the number of data included in each 2-hour periods were 400 800, depending on the season), to determine the relationship between the O2 and CO2 fluxes is preferable. The ORF
- values plotted in Fig. 8 show diurnal cycles with daytime minima in DJF, MAM and SON while no clear cycle is found in JJA. From 10:00 16:00 local time, the ORF values are in the range of 1.44 1.59 for all seasons. On the other hand, the ORF values from 18:00 9:00 local time are more variable, in the range of 1.39 1.74, and are clearly larger in the winter than in the summer.

The observed CO2 flux and the estimated O2 flux for each season are shown in Fig. 8. The CO2 flux shows clear diurnal

- 275 cycles with two peaks for all seasons, one in the morning and the other in the evening. The shape of the diurnal CO2 flux cycle, with larger flux in the winter than in the summer, was also found in our previous study at YYG for the period 2012-2013 (Hirano et al., 2015). On the other hand, the O2 flux shows similar diurnal cycles but in opposite phase with the CO2 flux. The daily mean CO2 fluxes were  $15.6 \pm 0.2$ ,  $11.2 \pm 0.1$ ,  $9.3 \pm 0.1$  and  $11.5 \pm 0.1$  µmol m-2s-1 in DJF, MAM, JJA and SON, respectively, while the respective daily mean O2 fluxes were  $-25.4 \pm 0.3$ ,  $-17.8 \pm 0.2$ ,  $-14.1\pm0.2$  and  $-17.7 \pm 0.2$  µmol
- 280 m-2s-1. The annual average daily mean O2 flux was -16.3  $\mu$ mol m-2s-1. 
[revised manuscript text omitted]
 O2 flux at YYG, calculated from the ORF and CO2 flux, was about -25 and -14 µmol m-2s-1 in the winter and the summer, respectively, which means the consumption rate of atmospheric O2 in an urban area of Tokyo is several hundred times larger than the global mean surface consumption rate. We estimated the average diurnal cycles of CO2 flux from the consumption of gas and liquid fuels for each season, based on

the average diurnal cycles of ORF and CO2 flux, and the CO2 emission inventory of human respiration around the YYG site.

- 365 Discrepancy between the estimated and inventory-based CO2 fluxes from gas fuels consumption was found to be the main cause of the significantly smaller evening peak of the observed total CO2 flux than that of the inventory-based total flux. Along with the peak in the estimated CO2 flux from the gas fuels consumption, the gradual increase in the estimated CO2 flux from the liquid fuels consumption found in the morning is consistent with the fact that the gas fuels consumption for domestic heating and cooking, and liquid fuels consumption from traffic during commuting occur in the morning. Therefore,
- 370 we can use simultaneous observations of ORF and CO2 flux as a powerful tool to validate CO2 emission inventories obtained from statistical data.

**Data availability.**

The data at YYG site presented in this study can be accessed by contacting the corresponding author.

375

**Author contributions.**

SI designed the study and drafted the manuscript. Measurements of O2 concentrations, CO2 concentrations, and CO2 flux were conducted by SI, SI and YT, and HS, respectively. HS prepared CO2 emission inventory data. NA prepared standard gas for the O2 measurements. SI and KT conducted O2 observations at MNM. HS, NK and HK examined the results and provided feedback on the manuscript. All the authors approved the final manuscript.

**380**

**Competing interests.**

The authors declare that they have no conflict of interest.

**385 Acknowledgements.**

 We thank Prof. T. Nakajima at Tokai University, Dr. Shohei Murayama and JANS Co. Ltd. for supporting the observation.

 This study was partly supported by the JSPS KAKENHI Grant Number 24241008, 15H02814 and 18K01129, and the

 Environment
 Research

 To be above the problem of the term of term of the term of term o

[revised manuscript text omitted]